# Parameter Optimization and Tuning Methodology for a Scalable E-Bus Fleet Simulation Framework: Verification Using Real-World Data from Case Studies

Mohammed Mahedi Hasan [1,2], Nikos Avramis [3], Mikaela Ranta [4], Mohamed El Baghdadi [1,2] and Omar Hegazy [1,2,*]

1   ETEC Department & MOBI Research Group, Vrije Universiteit Brussel, Pleinlaan 2, 1050 Brussels, Belgium
2   Flanders Make, 3001 Heverlee, Belgium
3   TNO Automotive, Automotive Campus 30, 5708 JZ Helmond, The Netherlands
4   VTT Technical Research Center, Vuorimiehentie 3, P.O. Box 1000, 02044 Espoo, Finland
*   Correspondence: omar.hegazy@vub.be; Tel.: +32-488-819-954

**Abstract:** This study presents the optimization and tuning of a simulation framework to improve its simulation accuracy while evaluating the energy utilization of electric buses under various mission scenarios. The simulation framework was developed using the low fidelity (Lo-Fi) model of the forward-facing electric bus (e-bus) powertrain to achieve the fast simulation speeds necessary for real-time fleet simulations. The measurement data required to verify the proper tuning of the simulation framework is provided by the bus original equipment manufacturers (OEMs) and taken from the various demonstrations of 12 m and 18 m buses in the cities of Barcelona, Gothenburg, and Osnabruck. We investigate the different methodologies applied for the tuning process, including empirical and optimization. In the empirical methodology, the standard driving cycles that have been used in previous studies to simulate various use case (UC) scenarios are replaced with actual driving cycles derived from measurement data from buses traversing their respective routes. The key outputs, including the energy requirements, total cost of ownership (TCO), and impact on the grid are statistically compared. In the optimization scenario, the assumptions for the various vehicle and mission parameters are tuned to increase the correlation between the simulation and measurement outputs (the battery SoC profile), for the given scenario input (the velocity profile). Improved simple optimization (iSOPT) was used to provide a superfast optimization process to tune the passenger load in the bus, cabin setpoint temperature, battery's age as relative capacity degradation (RCD), SoC cutoff point between constant current (CC) and constant voltage charging (CV), charge decay factor used in CV charging, charging power, and cutoff in initial velocity during braking for which regenerative braking is activated.

**Keywords:** e-bus powertrain; tuning and optimization; iSOPT; digital twins; internet-of-things





## 1. Introduction

Automotive system engineering has come a long way since Henry Ford spearheaded the assembly line process a century prior, resulting in sharp increases in productivity and manufacturing efficiency and corresponding decreases in the price of the manufactured vehicle [1]. The evolution in automotive system engineering in the 21st century saw the advent of Industry 4.0, empowered by the very high-speed internet (Internet 2.0), resulting in paradigm shifts in manufacturing production operations by merging the boundaries of the physical and virtual worlds [2]; the current state of the art (SotA) includes the Internet-of-things (IoT), cloud-connected processes, and digital twins (DT) technology. A DT model can have various levels of fidelity [3] in the virtual domain, but they are all tuned to accurately reflect a physical object or system. A DT model relies on the real-time measurement of data from numerous sensors installed in the physical system to continuously train itself to

behave as its physical counterpart to corresponding input stimuli [4]. A fully trained and tuned DT offers several advantages, including quicker iterative testing of the virtual model, using multiple copies for evaluation of different aspects of the vehicle at a fraction of the cost and time. For a city bus operator (CBO) and electricity distribution system operator (DSO), the virtual models can substitute for their real counterparts during fleet use case (UC) simulations to determine the real-world feasibility of electrification of the bus routes.

In [5], a simulation framework developed for the European Commission's Horizon 2020 project ASSURED was used to investigate the UCs of single buses and fleets of buses in various cities to determine their energy expenditure and TCO. The simulation framework was also used to study the reduction in energy utilization possible by applying different energy saving (ECO) strategies, and various optimization scenarios were investigated to determine the charging infrastructure that will minimize the fleet TCO (for the CBO) and load on the grid due to fleet charging (for the DSO). However, due to the lack of measurement data during the research conducted using various assumptions, including the use of a standard (hybrid SORT) driving cycle as the input scenario, constant average vehicle speed profile and a randomized passenger profile throughout the simulation period of one day, these assumptions naturally were not consistent with real-world conditions, including traffic situations on the road, and did not differentiate between peak and non-peak hours for passenger commutes. The hybrid SORT driving cycle can only be applied repetitively, synchronized to a constant average vehicle speed, throughout the simulation implying a constant traffic situation throughout the day. Furthermore, although the passenger profile was randomized, the output of the randomizer tended towards a full bus with time??, resulting in energy requirements that were aggressive. Similarly, the charging scenario assumed constant duration spacing in between two charging events, which resulted in a more simplified charging strategy. Finally, the results of the simulation framework were not validated using actual measurements; thus, the output of the simulation framework could only be taken as estimates.

In this research, the measurement data from the electric buses in the cities of Barcelona, Osnabruck, and Gothenburg are used to tune and validate the simulation framework. Furthermore, the study investigates the differences in energy consumption between the standard and actual driving scenarios, and finally an optimization was performed to determine the optimal charging strategy, given variable durations between two charging events, based on the input scenario. The objectives of this research are twofold: one is to validate the simulation framework, so that it can be used to investigate different scenarios with a high degree of confidence in its results; and two is to lay the framework for the creation of a DT of the electric bus for future research. Section 2 introduces the simulation framework and the necessary modification that enables it to work with actual measurements. Section 3 reports on the energy requirements from the vehicle demonstrations in cities. The tuning methodologies used to ensure that the output of the simulation framework matches the measured output, given similar inputs, are described in Section 4. Section 5 describes the optimization procedure for the charging strategy for bus fleets, whose driving scenarios were constructed using the actual driving scenarios. Finally, Section 6 concludes with how this research can be used to construct a DT from the simulation framework.

## 2. The Simulation Framework

A low fidelity (Lo-Fi) simulation framework illustrated in [5] was used to evaluate the energy expenditure for fleets of vehicles and impact on the electricity grid for a given mission profile in this research, with modifications in the framework to accept measurement data as the scenario input. An offline scenario input process was developed for the framework in this research, meaning that the simulation is not occurring parallelly in real-time using the measurement data taken during the bus demonstrations. Rather, the measurement data from the sensors are stored and later input to the simulation.

The simulation framework is based on basic electrical, mechanical, kinematic, and thermal equations needed to represent the charging infrastructure and forward-facing

electric bus (e-bus) powertrain model, as shown in Figure 1. Unlike a high fidelity (Hi-Fi) simulation model, where the simulation model is based on detailed physical equations of the actual system and uses small timesteps to ensure very high accuracy of the simulation output, the Lo-Fi framework uses look-up tables (LuTs) to define the efficiency maps of the various electronic, mechanical, electromechanical, and electrochemical devices integrated within the powertrain; and basic equations that model the overall energy transfer behavior of each component. The timestep in a Lo-Fi model is large to ensure high simulation speed at the cost of accuracy. Thus, even though a Lo-Fi model cannot simulate transient behaviors, they can be used to get a rapid estimate of the steady-state behavior. Therefore, Lo-Fi models can be used to simulate large time ranges covering the lifetime of the e-bus or large fleets of e-buses within a reasonable timeframe. Furthermore, a Lo-Fi model can be used to perform a fleet-level energy management and charging strategy (EM&CS) optimization, which require very fast simulation speeds. The Simulink framework was designed to use the measurement data from the demonstrations as inputs: the design of the energy storage system (ESS) block allowed comparison to be made between the simulated and measured battery SoC values for validation purposes, while the energy management system (EMS) block was designed to allow the model to be tuned to minimize the difference between the simulated and measured values. More details of the tuning process are provided in Section 4.

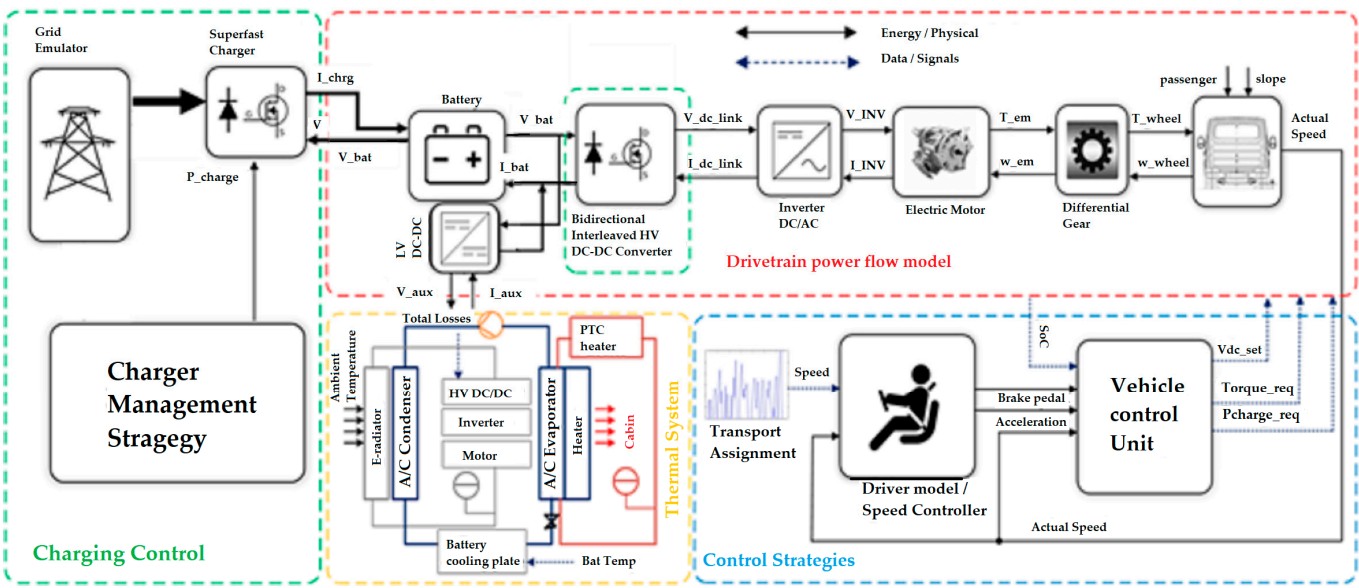

**Figure 1.** Overview of the simulation framework illustrating the forward-facing e-bus powertrain and grid infrastructure.

*Inputs to the Simulation Framework*

The framework was designed to accept measurement data from the bus as inputs in an offline process. In ASSURED, the various OEMs and CBOs involved in the demonstrations were responsible for the data collection process and then forwarding those data to the simulation team. However, different OEMs and CBOs used different data logging and data processing techniques. Therefore, it was not possible to apply a standardized methodology for data collection, making the offline validation the most suitable option. Table 1 gives a concise overview of the measurement data collected in each city. As can be seen, the collected data seems rather arbitrary; it is due to different stakeholders being involved in the data collection process. However, each stakeholder was required, at minimum, to provide the vehicle's speed profile (to be used as the simulation input) and battery SoC profile (to be compared with the simulation output), at a data logging frequency of 1 Hz, to ensure reasonable tuning and validation of the simulation framework. Beyond these constraints, each stakeholder communicated, according to their data sharing policies,

a subset of the following parameters: energy usage rate, mileage, charging state, charging time, ambient temperature, road inclination, GPS coordinates, and altitude.

**Table 1.** Overview of the measured data collected.

| City and Bus Type | Measured Parameter (Unit) | Logging Frequency |
|---|---|---|
| BCN, 12 m * | Speed profile (km/h)<br>Measured energy (kWh)<br>State of charge (%)<br>GPS coordinates (°) | 0.5 Hz |
| BCN, 18 m * | Speed profile (km/h)<br>State of charge (%) | 20 Hz |
| OSN, 12 m * | Speed profile (km/h)<br>Measured energy (kWh)<br>State of charge (%)<br>GPS coordinates (°) | 0.5 Hz |
| OSN, 18 m * | Speed profile (km/h)<br>Mileage (km)<br>State of charge (%)<br>Charging state (-) | 20 Hz |
| GOT, 12 m * | Speed profile (km/h)<br>State of charge (%)<br>Mileage (km)<br>Charging time (s)<br>Road inclination (°)<br>Ambient temperature (°C) | 10 Hz |

* BCN: Barcelona, OSN: Osnabruck, GOT: Gothenburg; 12 m and 18 m refers to the bus length.

Sensor data in vehicles are mainly communicated via the CAN bus network and logged via CAN-based dataloggers attached to the vehicle's CAN network and wirelessly communicated to a central server via the GSM (3G/4G) or Wi-Fi. The data is then decoded from the CAN message format (.blf), which is binary, into a more user readable format, including comma separated values (.csv), excel (.xlsx), or a simple text (.txt) file, using a CAN database (.dbc) file structure. The next step is to convert them into a common format, the MATLAB data (.mat) file, after which the parameter values are brought to a common sampling rate of 10 Hz, using up-and-down sampling techniques; wherein 10 Hz was chosen as a simulation time step of the Lo-Fi model. The data is then pre-processed to remove noise from the data, especially those which were measured via the GPS module, since GPS user accuracies, even with augmentation and when operated in wide open areas, are in "meters" for horizontal (i.e., longitude and latitude) measurements, and much worse for vertical (i.e., altitude) measurements [6]. In an urban setting featuring many obstacles (i.e., buildings, bridges etc.) and a multipath signal environment due to reflected signals, these accuracies are further degraded. Finally, the data is thoroughly checked to ensure that the speed and acceleration do not exceed the vehicle maximum for those parameters, and that the road inclination and difference in altitude between the lowest and highest point of the route were within known ranges.

## 3. Use Case Demonstration Overview

Numerous demonstration runs were conducted in the cities of Barcelona and Osnabruck using 12 m and 18 m e-buses, and in the city of Gothenburg using 12 m e-bus. For the 12 m bus, the demonstrations took place at two different months of the year to account for variation in weather. Table 2 provides the details for all the demonstrations considered for simulation and analysis. The simulations were run for approximately the same duration as their standard driving cycle counterparts in [5]; thus, some of the scenarios described in Table 2 were repeated until the desired timeframe was achieved. The complete specifications of the scenarios of the three routes, the 12 m and 18 m bus, as well as the climate

profile for each city, used as inputs for the simulations are presented in [5], while the exact maps of the demonstration routes are shown in the Appendix A.

**Table 2.** Overview of the demonstration scenarios.

| City and Bus Type | Demonstration Month | Operational Scenario | Route |
|---|---|---|---|
| BCN, 12 m * | December | 25.7 km in 160 min | H16 |
| | February | 16.7 km in 68 min<br>13.3 km in 31 min<br>26.3 km in 126 min | |
| BCN, 18 m * | June | 109.8 km in 558 min | |
| OSN, 12 m * | March | 49.1 km in 243 min<br>64.0 km in 310 min<br>63.2 km in 357 min | N5 |
| | May | 88.7 km in 473 min | |
| OSN, 18 m * | April | 88.1 km in 252 min | |
| GOT, 12 m * | May | 168.3 km in 784 min<br>99.5 km in 434 min | R55 |
| | October | 148.4 km in 575 min | |

* BCN: Barcelona, OSN: Osnabruck, GOT: Gothenburg; 12 m and 18 m refers to the bus length.

The measurement data gathered from the demonstrations were used to improve the UCs that were simulated using the standard driving cycles. Comparing the kinematic characteristics between the actual and standard driving cycles, very striking differences can be seen in their respective profiles. All measurements from the demonstrations exhibited accelerations whose ranges were higher than what was assumed when simulating the UCs using the standard driving cycle. Similarly, the maximum measured velocity from the demonstrations were higher than the maximum velocities assumed in the standard driving cycle, except in the case of the Osnabruck 12 m bus. Finally, in Barcelona, the average velocity measured during the demonstrations were higher than what was assumed in the standard driving cycle, while those of Osnabruck and Gothenburg were lower. From these facts, it can be assumed that the energy requirements for the buses subject to the measured driving cycles will be higher. Table 3 details the characteristics of the measured driving cycle from the demonstrations as well as the standard driving cycle, while Figure 2 illustrates this difference visually. As can be seen from the figure, the standard driving cycle is composed of clean and repeating patterns, while the actual measurements look random and somewhat noisy.

**Table 3.** Comparison between the demonstration and the standard driving profile characteristics.

| City and Bus Type | Demonstration Profile Characteristics | Standard Profile Characteristics |
|---|---|---|
| BCN, 12 m *<br>(4 demos) | Avg. vel. 9.65~26.2 km/h<br>Max. vel. 59.8~78.4 km/h<br>Max acc. 1.30~2.06 m/s$^2$ | BCN, Route H16, All buses:<br>Avg. vel. 9.52 km/h<br>Max. vel. 29.8 km/h<br>Max. acc. 0.51 m/s$^2$ |
| BCN, 18 m *<br>(1 demo) | Avg. vel. 11.8 km/h<br>Max. vel. 72.0 km/h<br>Max acc. 2.36 m/s$^2$ | |
| OSN, 12 m *<br>(4 demos) | Avg. vel. 10.6~12.8 km/h<br>Max. vel. 43.2~59.0 km/h<br>Max acc. 1.30~3.51 m/s$^2$ | OSN, Route N5, All buses:<br>Avg. vel. 19.8 km/h<br>Max. vel. 61.9 km/h<br>Max. acc. 1.06 m/s$^2$ |
| OSN, 18 m *<br>(1 demo) | Avg. vel. 21.0 km/h<br>Max. vel. 67.3 km/h<br>Max acc. 2.39 m/s$^2$ | |

Table 3. *Cont.*

| City and Bus Type | Demonstration Profile Characteristics | Standard Profile Characteristics |
|---|---|---|
| GOT, 12 m * (3 demos) | Avg. vel. 7.99~12.6 km/h Max. vel. 70.9~82.4 km/h Max acc. 4.99~5.33 m/s$^2$ | GOT, Route R55, 12 m bus: Avg. vel. 18.3 km/h Max. vel. 57.2 km/h Max. acc. 0.98 m/s$^2$ |

* BCN: Barcelona, OSN: Osnabruck, GOT: Gothenburg; 12 m and 18 m refers to the bus length.

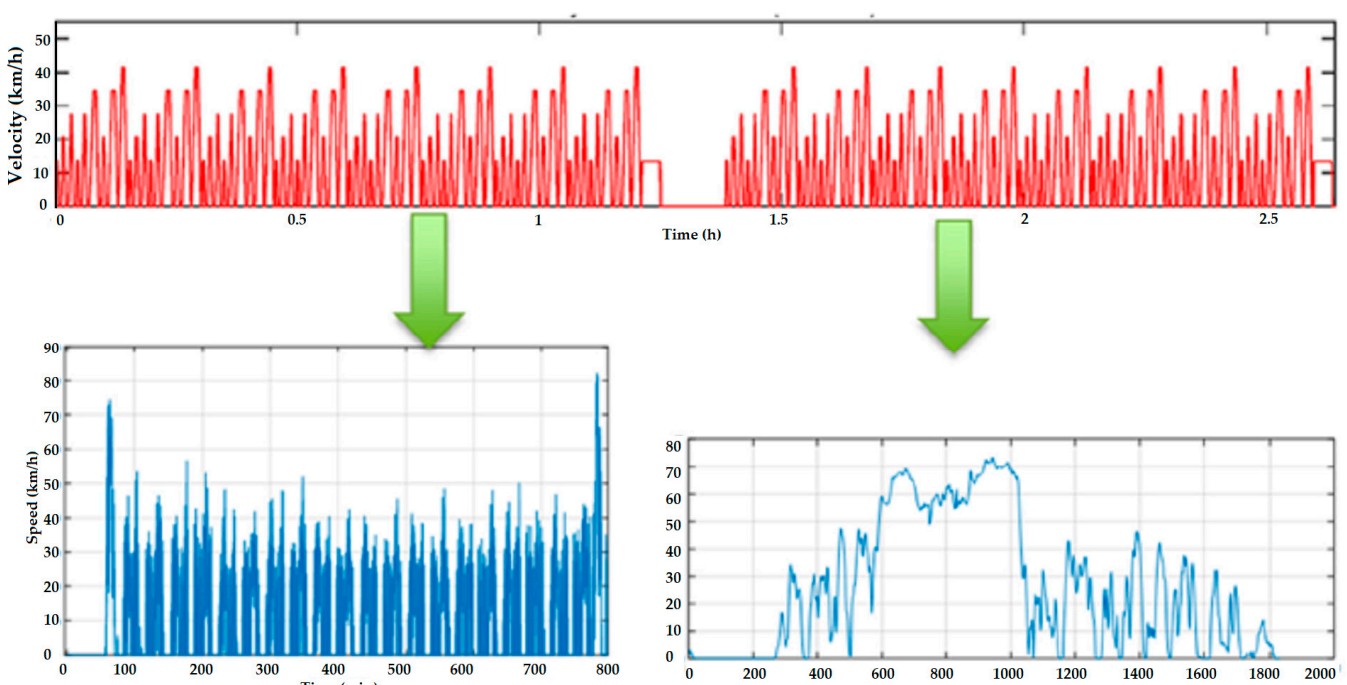

**Figure 2.** Comparison between the measured driving cycle from the demonstrations and standard driving cycles.

### 3.1. Simulation Output

Figure 3 compares the energy requirements determined from the simulation of the measurement data of the demonstrations to those from the UC simulations in [5], while Figure 4 illustrates the effects of the various ECO-features in reducing the energy consumption of the bus. For the remainder of the article, the baseline energy requirement is defined as the average energy requirement found from the UC simulations in [5] using the standard driving cycle. Figure 3 shows, as expected, that in Barcelona the energy requirements are significantly higher for the demonstrations compared with the baseline. However, the opposite is true for Osnabruck, where the energy requirements are significantly lesser than the baseline. This can be explained by the simple fact that in Barcelona, the average and maximum speeds of the buses in the demonstration are much higher than the baseline. Thus, the buses in the demonstration experience higher aerodynamic drag, leading to greater energy requirements compared with the baseline. In the case of Osnabruck, the opposite was true; for the 12 m bus, the average and maximum speeds of the demonstrations were less than those of the baseline, thus lesser energy was required than for the baseline. In the case of the 18 m bus, the average and maximum speeds are comparable between the demonstrations and the baseline; thus, the energy requirement between the baseline and demonstration is similar. For Gothenburg, the average velocity is less than the average velocity of the baseline, even if the maximum velocity is higher. Thus, the bus expends less energy on average compared to the baseline. From the results, it can also be deduced that normal acceleration and deceleration have a low impact on the rate of energy expenditure of the vehicle; this can be explained by the fact that the vehicle is an electric

bus with an efficient energy recovery system (via regenerative braking), thus 70% to 80% of the traction energy expended during acceleration is recovered during braking [7]. The amount of energy recovered depends on several factors including the momentum of the vehicle during braking, SoC of the battery, and capability of the battery to accept the power influx. For small EVs such as the Renault Zoe, the cutoff velocity beyond which energy recovery can efficiently occur during braking is 5 m/s [7], but for heavy-duty vehicles such as buses, the regeneration can occur from a lower velocity due to their larger masses resulting in greater braking momentum. Thus, regenerative braking in an urban scenario with low speeds and heavy traffic is more suitable for electric buses and trucks.

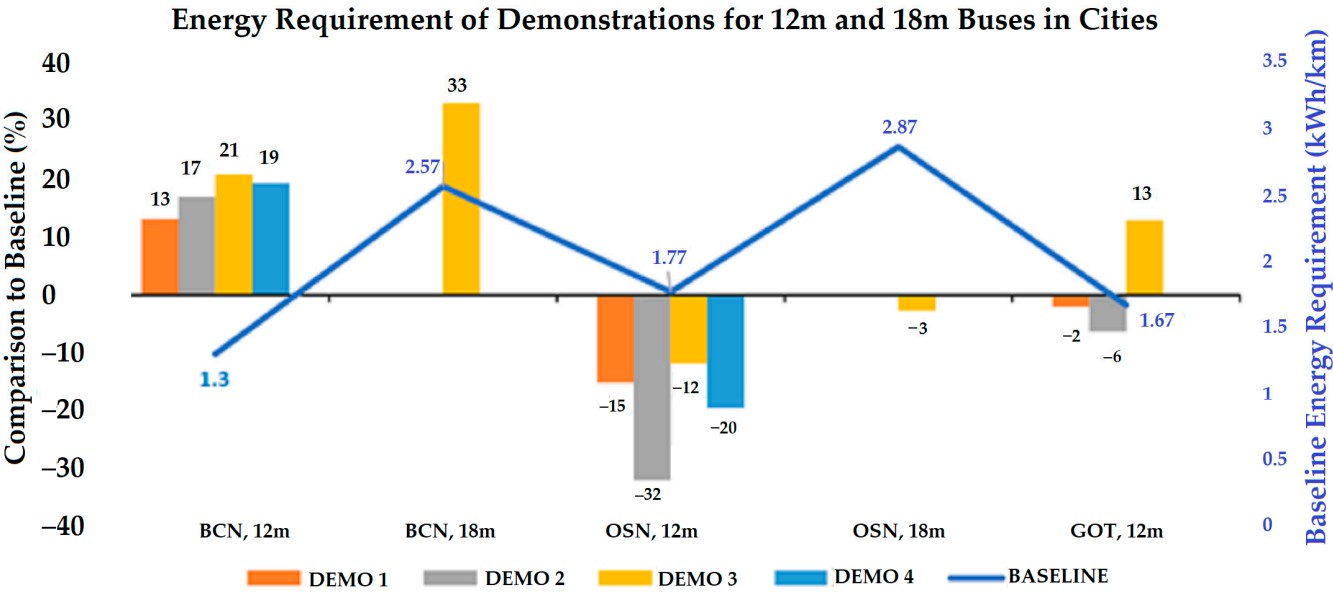

**Figure 3.** Comparison of the energy requirement for 12 m and 18 m buses in Barcelona, Osnabruck, and Gothenburg between the standard and actual driving cycles.

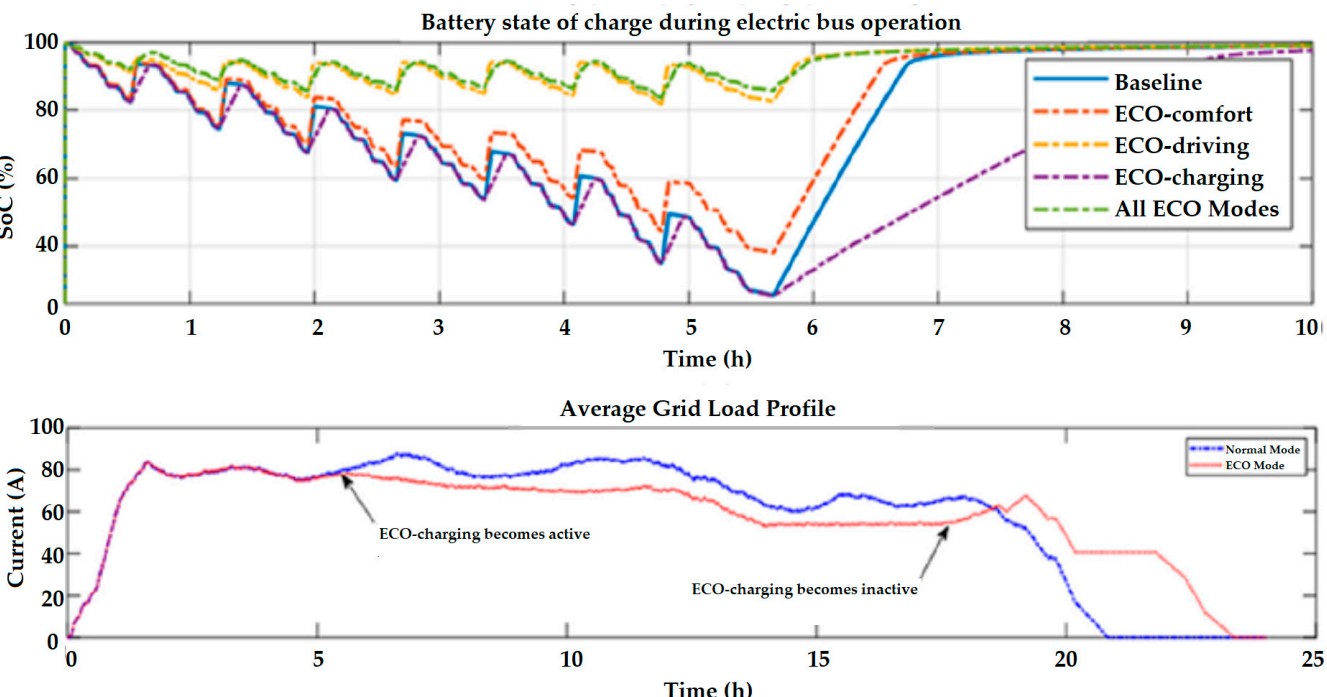

**Figure 4.** Effects of the ECO-features of the energy requirements (**top row**) and impact on the electricity grid (**bottom row**).

*3.2. Energy Reduction Using ECO-Features*

Three energy management techniques are considered to reduce the energy requirements of the buses, namely, ECO-comfort [8], ECO-driving [9], and ECO-charging [10,11]. ECO-comfort optimizes the thermal management system of the bus responsible for the cabin and battery cooling systems, ECO-driving optimizes the EMS of the bus responsible for vehicle traction and regeneration, and ECO-charging optimizes the charging management system of the vehicle responsible for battery charging. Figure 4 highlights the effects of the ECO-features on the SoC; and details about the functionality of the three ECO-algorithms are presented in the Appendix B. Based on the SoC profile shown in the top row of Figure 4, ECO-driving has a significant effect on energy reduction, as seen from the smaller drop in the battery SoC with ECO-driving compared with the baseline. This is because the baseline driving profile featured aggressive driving, i.e., high speed (max. velocity of 18.7 m/s) and acceleration (max. acceleration of 2.39 m/s$^2$), and ??these see the highest reduction in the energy requirement due to the application of ECO-driving. There is modest energy savings due to ECO-comfort, as it was simulated for moderate springtime weather conditions. ECO-charging does not change the energy requirement of the vehicle compared to the baseline, but as can be seen from the bottom row of Figure 4, it does spread out the charging duration, resulting in a lower average load on the electricity grid; this is important during fleet charging so as not to put undue stress on the electricity grid. Overall, the 12 m bus saw an average reduction of 0.4 kWh/km from the baseline energy requirements, while the 18 m bus had a reduction of almost 1.8 kWh/km from the baseline. On average, at least three quarters of the reduction was achieved due to ECO-driving, while barely 2% is due to ECO-charging, as shown in Figure 5.

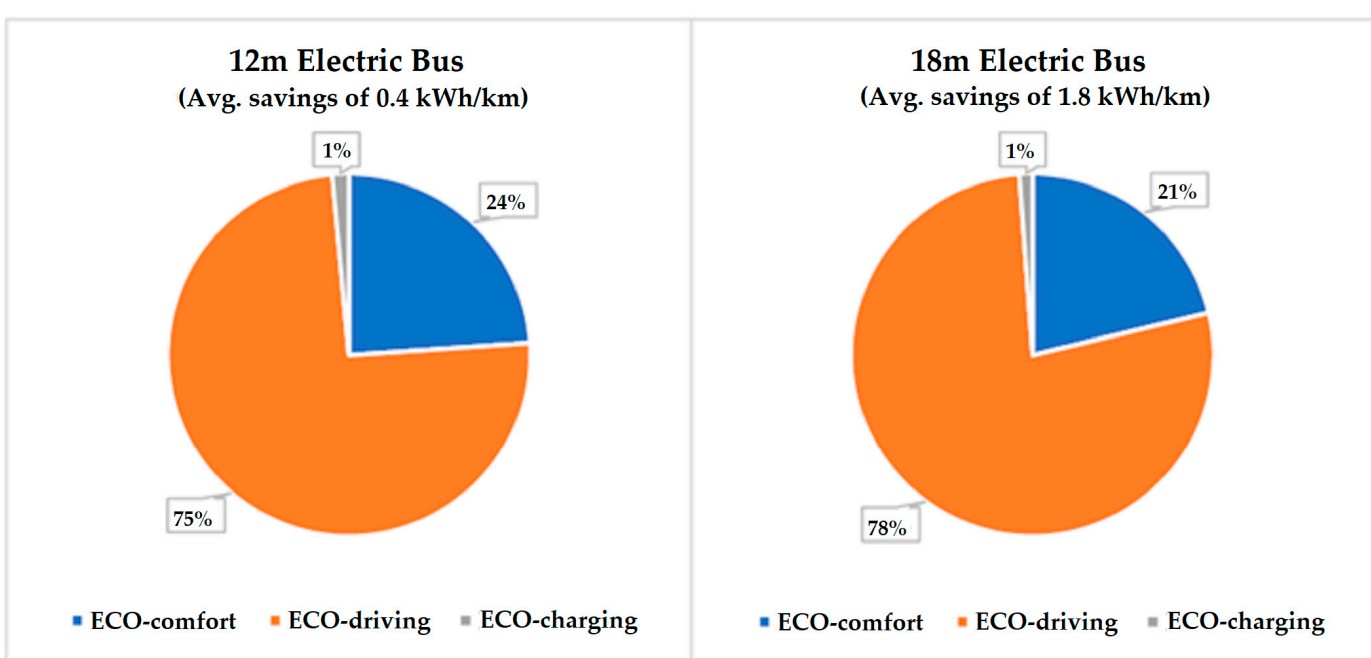

**Figure 5.** Breakdown of energy savings due to ECO-features for 12 m and 18 m electric bus.

Table 4 shows that there is a high correlation between the amount of energy savings due to ECO-driving and average speed of the vehicle in the baseline scenario. There is also a link between the size of the bus and possible energy savings. However, the data also show that there is no link between the top speed of the vehicle in the baseline scenario and possible energy reduction; this may be because the vehicle does not spend sufficient time at its top speed for it to matter. These results prove that there is a lot of room for improvement when it comes to driving behavior and low-speed driving is recommended for optimum traction energy utilization.

**Table 4.** Energy reduction possible due to ECO-driving.

| City, Bus Type and Demo Number | Speed (Mean and maximum) | Baseline Energy Requirement | Energy Savings |
|---|---|---|---|
| BCN 12 m, Demo 1 * | 2.68 m/s, 21.9 m/s | 1.47 kWh/km | 21.8% |
| BCN 12 m, Demo 2 * | 4.08 m/s, 16.6 m/s | 1.52 kWh/km | 22.8% |
| BCN 12 m, Demo 3 * | 7.28 m/s, 20.4 m/s | 1.57 kWh/km | 27.5% |
| BCN 12 m, Demo 4 * | 3.48 m/s, 21.4 m/s | 1.50 kWh/km | 20.9% |
| BCN 18 m, Demo 1 * | 4.28 m/s, 20.0 m/s | 3.42 kWh/km | 39.5% |
| OSN 12 m, Demo 1 * | 3.37 m/s, 16.0 m/s | 1.50 kWh/km | 24.2% |
| OSN 12 m, Demo 2 * | 3.44 m/s, 12.0 m/s | 1.20 kWh/km | 17.6% |
| OSN 12 m, Demo 3 * | 2.95 m/s, 16.4 m/s | 1.56 kWh/km | 21.9% |
| OSN 12 m, Demo 4 * | 3.13 m/s, 14.7 m/s | 1.42 kWh/km | 18.7% |
| OSN 18 m, Demo 1 * | 5.82 m/s, 18.7 m/s | 2.79 kWh/km | 47.4% |
| GOT 12 m, Demo 1 * | 3.50 m/s, 22.9 m/s | 1.83 kWh/km | 7.4% |
| GOT 12 m, Demo 2 * | 2.22 m/s, 21.6 m/s | 1.75 kWh/km | 8.1% |
| GOT 12 m, Demo 3 * | 3.30 m/s, 19.7 m/s | 2.11 kWh/km | 5.3% |

* BCN: Barcelona, OSN: Osnabruck, GOT: Gothenburg; 12 m and 18 m refers to the bus length.

## 4. Validation of the Simulation Framework

This section focuses on the methodology followed to validate the simulation framework through real measurement data from the demonstrations. The measurements were also used to improve the inputs to the simulation model to have a better representation of the UCs; these improved inputs are then used for the simulation. Measurement data from Osnabruck and Gothenburg were used in the validation process. The quality of the data from the two sources were different. The Gothenburg dataset consists of continuous measurement values sampled at 20 Hz directly from the vehicle's CAN-bus. The data from Osnabruck, extracted from the CBO's cloud server, were only available at intermittent intervals. Thus, the two cases were handled differently.

The validation and tuning process addressed the following features:

- The EMS: The energy recovery system was tuned to align the traction energy profile with the measurement data. The regenerative braking system (RBS) is a proprietary system for many OEMs; thus, assumptions were made during model development.
- The charging management system (CMS): The cutoff between the constant current (CC) mode and the constant voltage (CV) mode, and the current decay parameter during the CV mode were tuned based on the measurement data. These parameter values are also not forthcoming by the OEMs.
- The passenger load estimation: Passenger load inside the bus is the one aspect that could not be automatically measured and requires manual counting; thus, it is usually ignored. Instead, some simulations involved an intricate passenger model based on the passenger appearance rate at each bus stop as a function of time [12], which is modeled on actual bus traffic data by the CBO. Others use agent-based modeling whereby each passenger is a unique object that has "preferences", such as drop in point, drop off point, and waiting time [13]. In [14], a cellular automata model is utilized to study behavioral characteristics of bus passengers boarding and alighting behavior. There are also certain cases where a fixed load was assumed within the bus, based on passenger load factor [15], when the passenger load is ancillary to other considerations. The UC simulations carried out in [5] assumed a random passenger profile as a function of time within the bus cabin; however, for this validation, the passenger inside the bus was estimated based on the measured SoC profile.

### 4.1. Tuning and Validation Methodology

The tuning was performed by using optimization to directly determine the parameters' values of the powertrain module (e.g., EMS, CMS, BMS) that needs to be tuned, to

minimize the normalized root mean squared error (NRMSE) between the simulated and measured outputs.

$$C_{total} = \frac{\sqrt{\frac{\sum_{i=1}^{n}(SOC_{SIM,\ i} - SOC_{MEAS,i})^2}{n}}}{\max(SOC_{MEAS}) - \min(SOC_{MEAS})} + constraint\ penalty \tag{1}$$

The cost function, $C_{total}$, shown in (1), gives an estimate of the deviation between the simulated and measured SoC of the battery. The closer the value of the cost function is to zero, the closer the match between the two SoC signals. To achieve a minimum value of $C_{total}$ in the optimization process, not only must the two SoC signals match as closely as possible, but the simulation must also not violate any of the constraints elaborated on in Section 4.2. The output score calculated by the NRMSE ranges from 0 (perfect match between simulated and measured signals) to 1 (implying the maximum mismatch between the two signals), thus any penalty applied has values greater than 1. The magnitude of the penalty depends on the extent of the violation of a given constraint.

By the standard definition [16], the tuning methodology described in this study is an example of the offline tuning process because the tuning occurs in the simulation model using saved data, i.e., the measured input during the demonstration was not processed in real-time but cached for later processing and simulation. Instead, for this study, a different definition is used to differentiate between an offline and an online tuning process. The online tuning process is defined as the tuning that occurred while the simulation was still ongoing, whereas in offline tuning, the tuning occurred in an iterative process between separate simulations, after each simulation had finished running in its entirety. For the online tuning process, the total time duration of the simulation was split into several "windows"; the tuning occurred in between each time window, and its result was applied to the next window until no further improvement could be seen, i.e., it converged. If the convergence occurred before the end of the simulation, the tuning was considered completed; otherwise, the simulation was repeated with the latest tuned configuration as the starting condition. As expected, the online process is faster due to the small dataset involved in the tuning process, so the tuning completes quicker.

The online tuning process was applied during the optimization; the simulation time duration was split into variable-sized windows based on the driving cycle. As there were no discernable patterns, the split was made according to different categories of driving, such as constant speed driving or driving with frequent accelerations, as shown in Figure 6. The tuning algorithm assessed various parametric configurations within a window sample to minimize the NRMSE within that window, before applying the best possible configuration to the next window and repeating the process, as in [16].

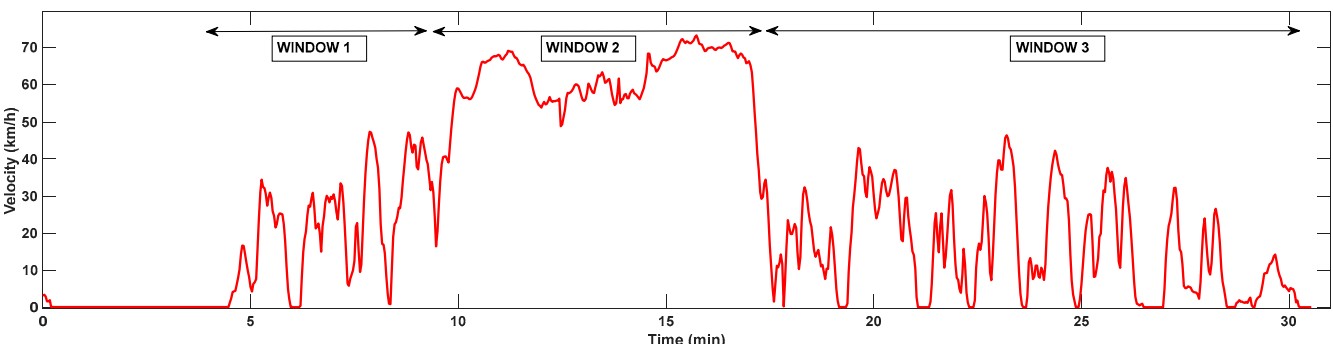

**Figure 6.** Tuning the simulation using variably sized windows applied based on the driving cycle.

### 4.2. Optimization Based Tuning Process

In [17], constrained minimization was used in the tuning process of the controller to allow the controller to become flexible, so it can respond in a robust fashion to changes in the inputs, and be used for different purposes by optimally retuning the control parameters

subject to different constraints. In [18], a constrained nonlinear optimization was carried out using a sequential quadratic programming (SQP) algorithm to tune PID gains to allow the controller to adapt to changes in the plant; this not only offered superior performances when compared to traditional PID tuning, the tuning process was much quicker. Similarly, linear programming was utilized in [19] to tune the weights of a symmetric finite impulse response (FIR) filter of low-bandwidth controllers for a linear time and spatial invariant (LTSI) systems; a hybrid genetic algorithm (GA) followed by constrained nonlinear minimization was used in [20] to optimize in real time the autopilot gain of an unmanned aerial vehicle (UAV); the GA ensured a global minimum, but without running the GA process to its conclusion, and the $f_{mincon}$ function utilized to finetune the results of the GA at a higher speed. In this study, the meta-heuristic algorithm, improved simple optimization (iSOPT) [21], was used to tune the EMS and CMS of the electric bus powertrain model, to ensure a global minimum within the fastest possible time, so that the tuning can be carried out in real-time.

The set of parameters that were tuned for the EMS are:

- Cutoff velocity for regenerative braking activation
- Passenger load in the bus (broad categories: full load, half load, driver only)

The set of parameters that were tuned for the thermal management system (TMS) are:

- Cabin setpoint temperature

The set of parameters that were tuned for the CMS are:

- Cutoff SoC between CC and CV charging mode
- The current decay factor for CV charging mode
- The charging duration and power
- Initial Battery ageing

The final two parameters that were tuned are the passenger load estimate in the bus and the cabin setpoint temperature. Thus, a total of seven parameters makes up the solution space. An initial population size of 11 with random combinations of the seven parametric values was generated, and the algorithm described in [5] is followed till its conclusion. The maximum number of iterations was set to 50. The optimization is handled via MATLAB scripts, which populates the variables of the Simulink model with updated values every iteration while simulating the demonstration scenario.

The following constraints were applied to the optimization, and a penalty was added to the optimization score if one or more of these constraints were exceeded in any way:

- The current decay factor, cutoff velocity, and cutoff SoC were positive
- The cutoff SoC was below 100%
- The RCD was below 25%
- The charging duration exceeds 1 min and charging power was positive
- The battery SoC should not drop below 10% during the simulation

The advantage of using optimization techniques to tune the model is that it preserves the integrity of the model, with the only factor being changed is the set of parameter values of the respective modules that are being tuned. The improvements of the optimization methodology followed in this research compared with [5] are twofold. The first is an improvement in speed of optimization. In all cases, it is noticed that $T_{opt} < n * T_{sim}$, where n was the number of iterative simulations required during the optimization before convergence and $T_{sim}$ is the duration of one complete simulation. This is because using the methodology in [5], we would have needed to run the complete simulation 'n' times before convergence, but with the window technique presented in this article, we only needed to run the 1st and the 2nd windows 'n' times, and the subsequent windows needed to be run less than 'n' times, as the parameters values have already become optimal by that point. The second improvement was the fact that the optimization process could be made online in the traditional sense [16] by focusing on optimizing the model using the measurement data dump from a previous time window, while the measurement is in progress for the

current time window. This is a necessary first step to overcome in the process to develop a real-time DT of the system, which is the end goal of this research track.

## 5. Validation Results

### 5.1. Osnabruck

The demonstration for Osnabruck city took place in the months of March, April, and May using an 18 m bus type. Measurement data are available for a total of 9 days, with 2 days each in March and May, and the rest in April. The demonstrations focus on different charging characteristics, with the March and April demonstration clearly focusing on low-power depot charging, and the May demonstration focusing on the high-power opportunity charging. The measurement data provided included the time, speed, and distance travelled data taken at 5-minute intervals. The sampling rate of the provided data is not sufficient to perform simulation and, therefore, each five-minute interval was replaced by the standard SORT driving cycle whose mean velocity was adjusted to match the measured speed value if the distance covered by the adjusted driving cycle was less than or equal to the actual distance traversed during that five-minute interval. If, on the other hand, the adjusted driving cycle covered a larger distance than the actual measured value, the simulation was conducted assuming a constant velocity for that five-minute interval. The measurement also consisted of the battery SoC level at different points during the demonstration. These SoC values are used to verify the simulation results by comparing the simulated SoC values with the actual demonstration SoC values at the same point in time. The simulation assumptions were tuned to give the scenario configuration that provides a simulation with the closest match between the simulated SoC values and measured SoC values.

Figure 7 illustrates the driving and charging scenario constructed from the demonstration data provided for March 29th and 31st, April 7th, 12th & 13th, and 20th & 21st, and May 6th and 12th. The driving and charging scenario will be shown within the same plot. There is no charging taking place between the 20th and 21st; the vehicle is switched off and restarted the next day. The estimated (average) power of the charger used during the March and April demonstrations is 18 kW, thus making it an AC charger in the depot; the estimated power of the charger used for the May demonstrations is 290 kW, thus making it a DC fast charger used for opportunity charging. The charging duration is determined by the type of charger, with opportunity charging active for 10 minutes, while the depot charging is active for hours. The total increase in battery SoC during charging was used to estimate the rated power of the charger.

Table 5 lists the estimates of the driving scenario that gave the closest SoC match between the simulated values and demonstration measurements. Based on these estimates, Figure 8 shows the validation output of the simulation framework.

**Table 5.** Estimation of the driving scenario configuration for the Osnabruck demonstration.

| Parameter to Be Estimated | March | April | May 6th | May 12th |
|---|---|---|---|---|
| Passenger load | Driver only | Full load | Full load followed by driver only | |
| Cabin setpoint temperature | 20 °C | 15 °C | | 20 °C |
| Charging type | Depot | | Opportunity | |
| Charging power | 18 kW | | 290 kW | |
| Battery capacity | 120 kWh | | | |
| Initial battery age | Relative capacity degradation of 20% | | | |
| RBS cutoff velocity | RBS active when vehicle speed above 1.5 m/s | | | |

### 5.2. Gothenburg

The demonstration for Gothenburg city took place in the month of May and October using 12 m bus. Measurement data are available for a total of 3 days, with 2 days in May and 1 day in October. The demonstrations focus on different charging characteristics, with the May demonstration clearly focusing on shorter duration opportunity charging in the constant voltage (CV) mode, and the October demonstration focusing on the longer duration opportunity charging in the constant current (CC) mode. The duration of the May demonstration was between 12 h to 14 h per day, while the October demonstration was limited to below 3 h. The Gothenburg demonstration had access to continuous driving cycle data; thus, the actual speed measurements were used as inputs after suitable preprocessing. Furthermore, the Gothenburg demonstration also had access to the road inclination profile and ambient temperature profile, in addition to the velocity profile, as inputs. Thus, more relevant simulations could be produced for the validation process. The speed tracking and battery SoC level were validated by comparing the measured values against the simulated values. Table 6 lists the estimates made for the simulations, which achieved a high correlation between the simulated and measured SoC.

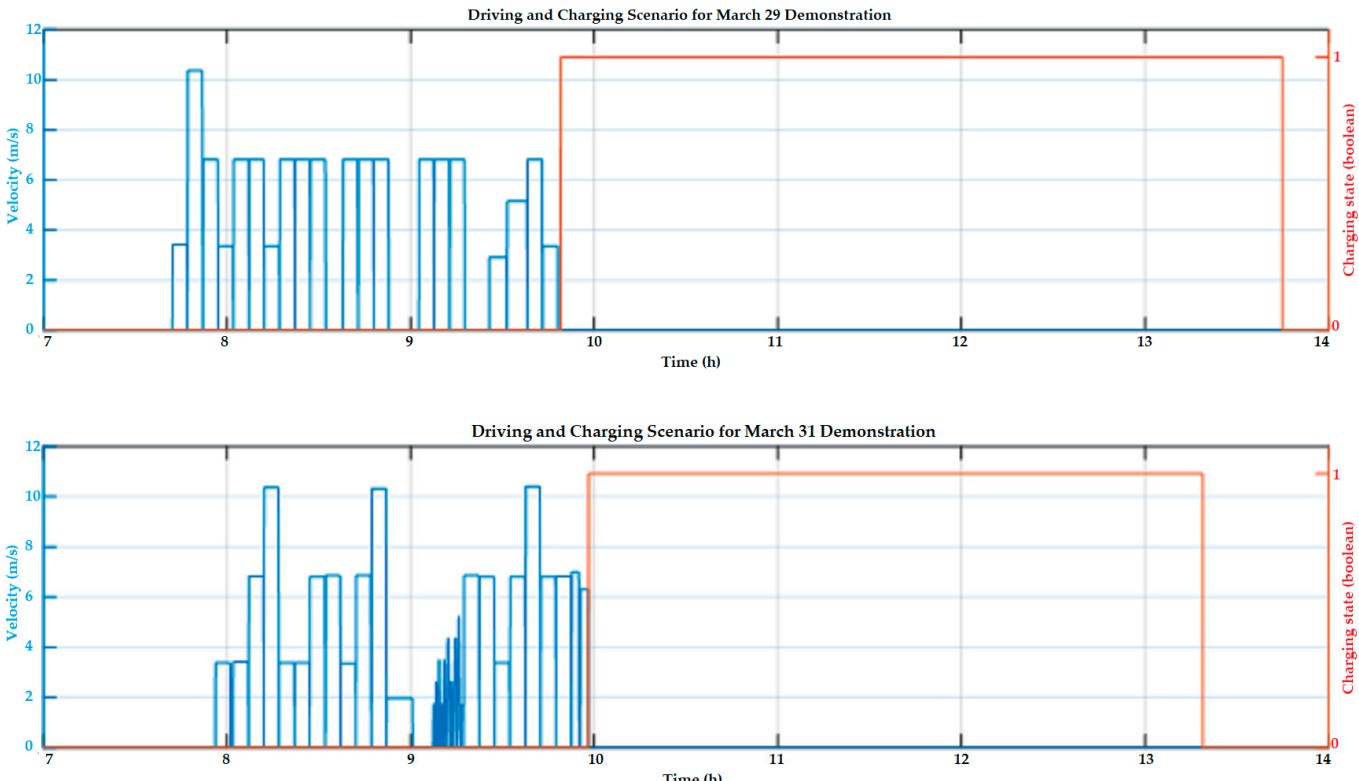

**Figure 7.** *Cont.*

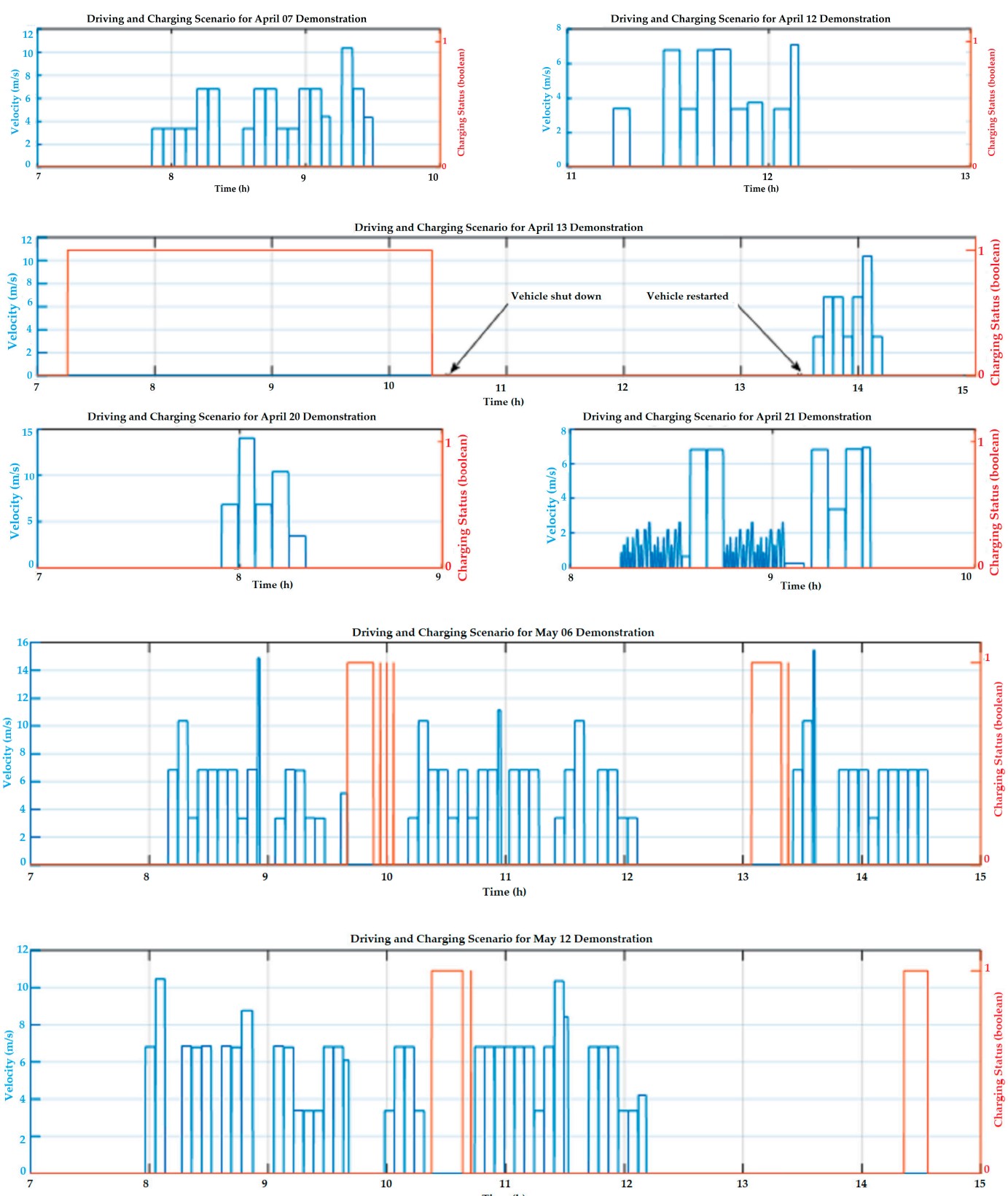

**Figure 7.** Osnabruck driving scenarios during the 18 m electric bus demonstrations.

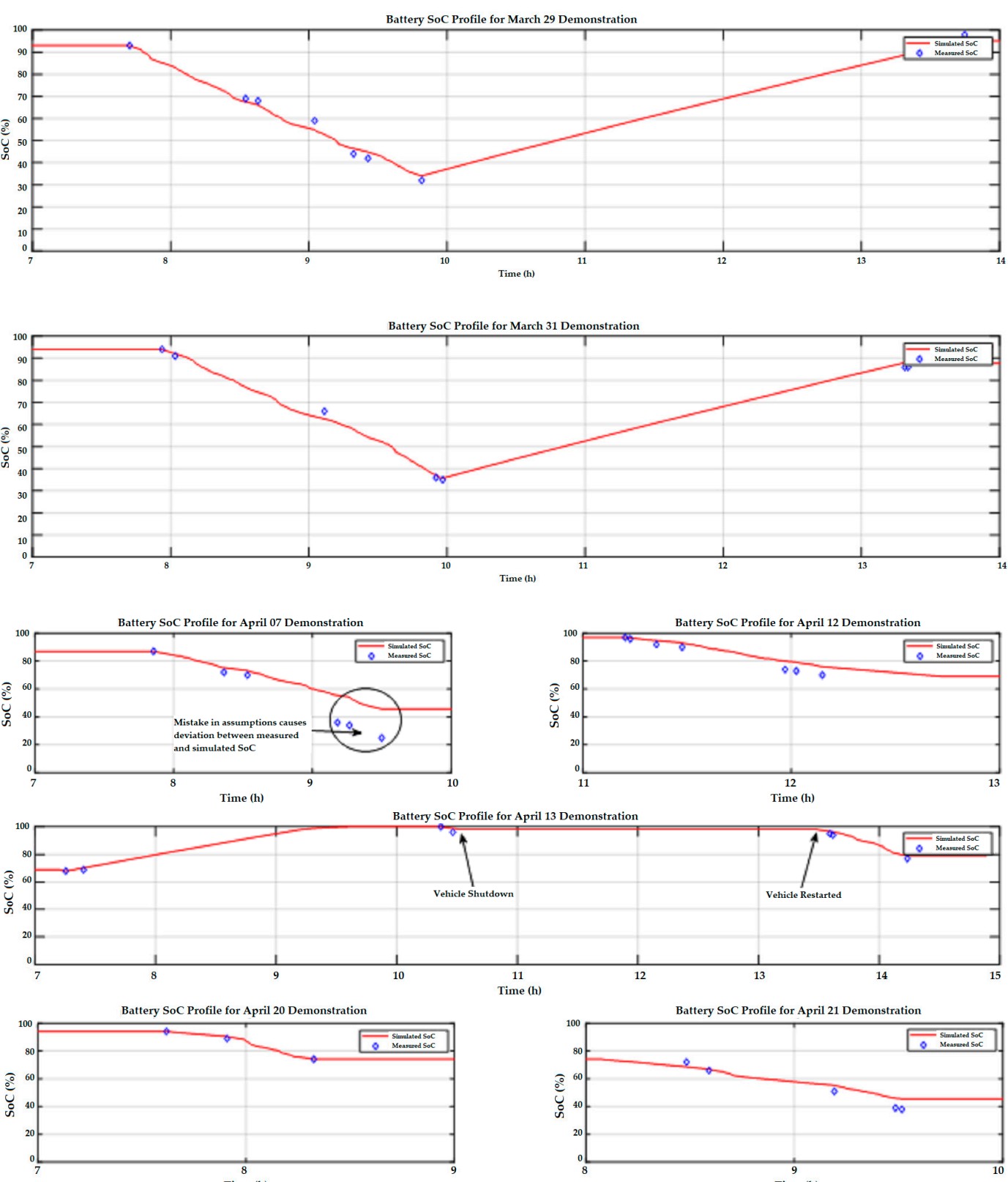

**Figure 8.** *Cont.*

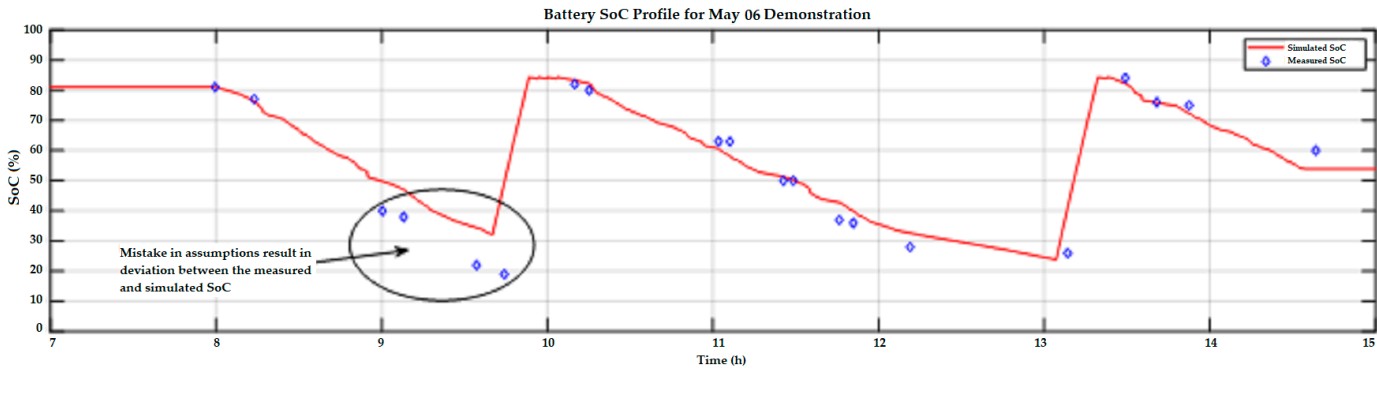

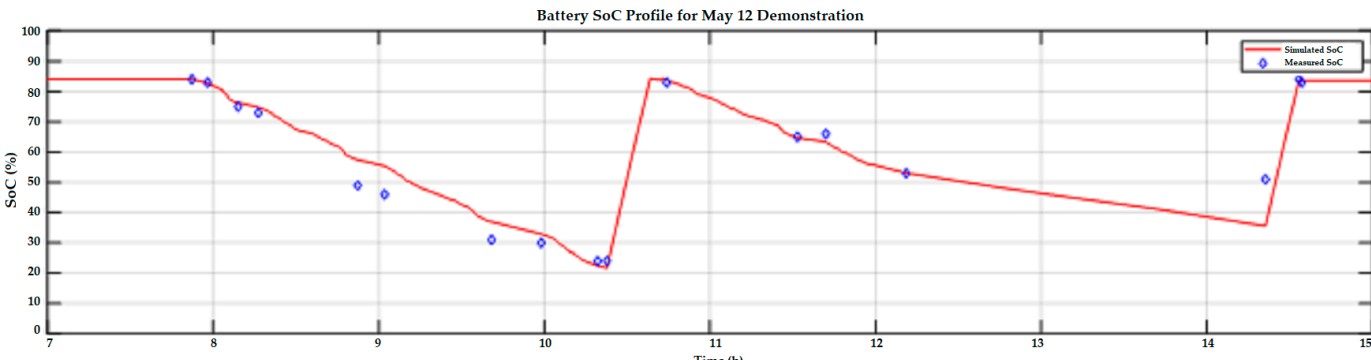

**Figure 8.** The SoC output profile of the 18m bus subject to the driving and charging scenario in Figure 7 using the estimates given in Table 5, and showing the correlation between the simulated outputs and measured values.

**Table 6.** Estimation of the driving scenario configuration for Gothenburg demonstration.

| Parameter to Be Estimated | March | April | May 12th |
|---|---|---|---|
| Passenger load | Only driver initially, full load between 1 h and 4 h, then half load until 13 h, then driver only until end | Driver only | Only driver when idle (i.e., at end of the route or during charging), full load when bus is moving |
| Cabin setpoint temperature | 20 °C | | |
| Charging type | Opportunity | | |
| Charging power | 450 kW (current decay has a $\beta = 0.23$ in CV mode, which is activated when SoC > 87.5%) | | |
| Battery capacity | 200 kWh | | |
| Initial battery age | New batteries with no degradation | | |
| RBS cutoff velocity | RBS active when vehicle speed above 1.5 m/s | | |

Figure 9 shows the results of the October 13th demonstration based on the assumptions listed in Table 6. The total demonstration was conducted over 2.5 h with the bus standing idle for the first 40 min. The bus charged using an opportunity charger, with a rated power of 450 kW, at the 1.5 h mark. The simulation tracks the speed accurately with minimal deviation between the simulated and measured outputs. The battery SoC is also tracked accurately; however, there is a deviation between the measured and simulated outputs when the bus is standing idle. The energy usage during that time is very high according to the measured SoC values, which cannot be reasonably explained, unless the speed signal is missing/corrupted.

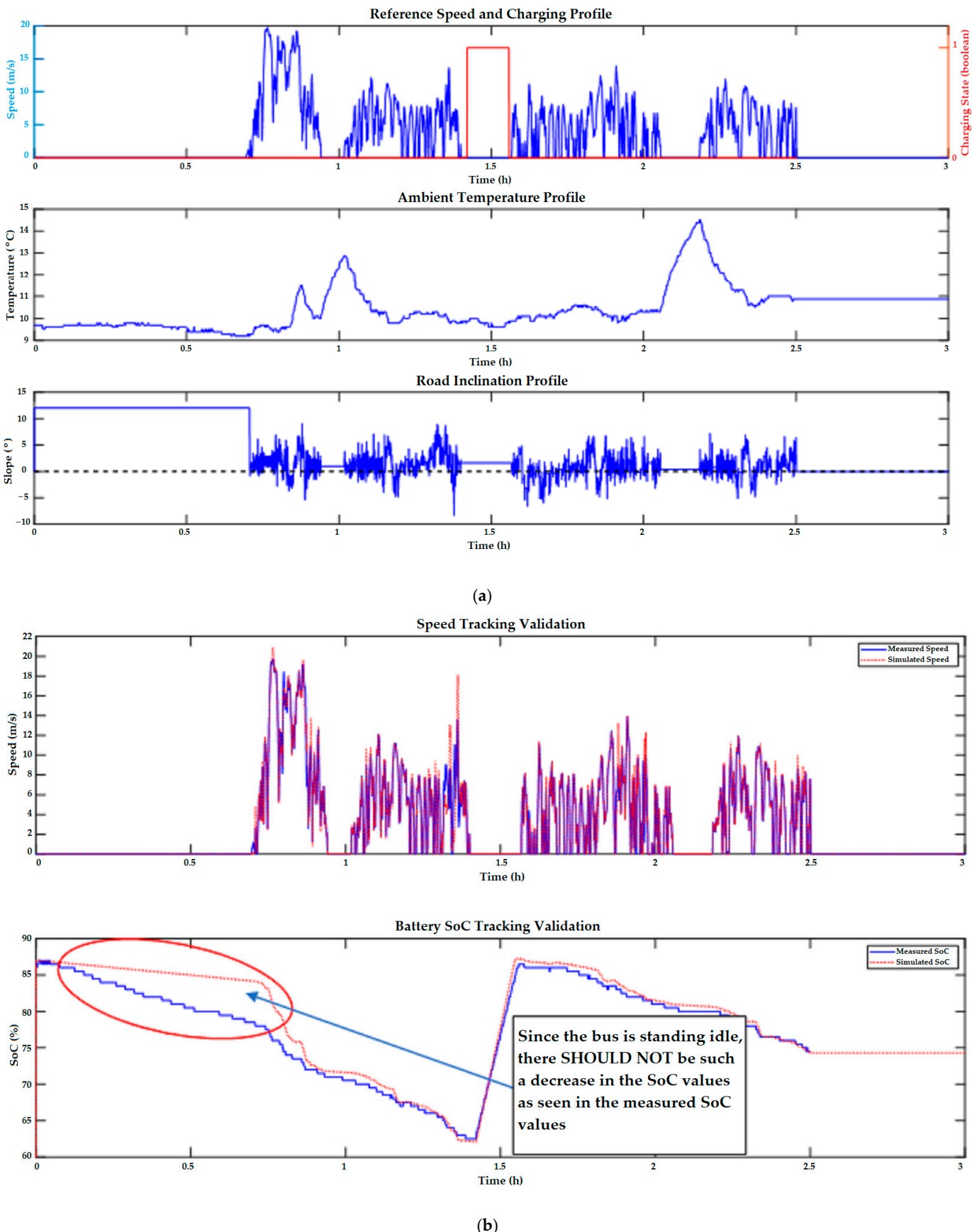

(**a**)

(**b**)

**Figure 9.** Validation of the October 13th demonstration of 12 m bus in Gothenburg city using the estimates in Table 6. (**a**) Scenario inputs, (**b**) Scenario outputs and validation.

Figure 10 shows the results of the May 27th demonstration based on the assumptions listed in Table 6. The total demonstration was conducted over 13.5 h with the bus standing idle for the first 1 h. The bus was charged using an opportunity charger, with a rated power of 450 kW at 22 different instances. The first charging instance occurs entirely in the CC mode and the second charging instance happens partially in both CC and CV modes, while the remaining charging occurs entirely in the CV mode. The charging current decay (β) of 0.23, when battery SoC exceeds 90%, accurately models the measured charging current. There is a deviation between the measured SoC and simulated SoC at two points; one when the bus was standing idle and the measured SoC showed greater than expected energy usage for an idle vehicle, and the other when the reference speed of the bus was 83 km/h, which exceeded the modeled maximum speed of the bus of 80 km/h.

Figure 11 shows the results of the May 29th demonstration based on the assumptions listed in Table 6. The total demonstration was conducted over 12.5 h; however, the measurements are only available after the 5 h mark. The bus was charged using an opportunity charger, with a rated power of 450 kW at 14 different locations; all the charging events were short in duration. For this demonstration, all charging events occurred entirely in the CV mode. Unlike the other demonstrations, which were modeled with high passenger loads, this one is modeled with only the driver to account for the minimal energy utilization observed. There is a deviation between the measured SoC and simulated SoC at a few locations; the deviations are most likely due to inaccurate battery models for LFP battery chemistry above 90% SoC. The deviations in the beginning can be explained by the fact that the measurements prior to the 5 h mark are not presented; thus, it was not possible to determine the state of the bus prior to the start of the simulation. The deviation at the end was most likely due to a more efficient energy recovery process during regenerative braking than was accounted for in the vehicle model.

There is a deviation between the measured SoC and simulated SoC at a few locations in Figure 11; the deviations are most likely due to inaccurate battery models for LFP battery chemistry above 90% SoC. The deviations in the beginning can be explained by the fact that the measurements prior to the 5 h mark are not presented; thus, it was not possible to determine the state of the bus prior to the start of the simulation. The deviation at the end was most likely due to a more efficient energy recovery process during regenerative braking than was accounted for in the vehicle model.

One of the clear outcomes of the validation process was an accurate determination of the current decay factor (β) during the CV mode of charging and the CC/CV cutoff SoC value. It is understood that after the bulk charging phase of a battery in CC mode, the charging switches to the CV mode, where the current reduces to a trickle. This reduction of the current was modeled as an exponential decay once the battery SoC exceeds 87.5%.

The decay amount is given as:

$$\text{Iout} = \begin{vmatrix} I_{max\_cc} \times e^{\beta \times (SoC-90)}, & SoC > 87.5 \\ I_{max\_cc}, & \text{otherwise} \end{vmatrix} \tag{2}$$

where $I_{max\_cc}$ is the maximum charging c-rate during the CC charging mode (-3C for an LFP battery chemistry), and β is the decay factor; it was found that a β of 0.23 models the charging current that gives the closest correlation between the measured and simulated battery SoC profile during charging.

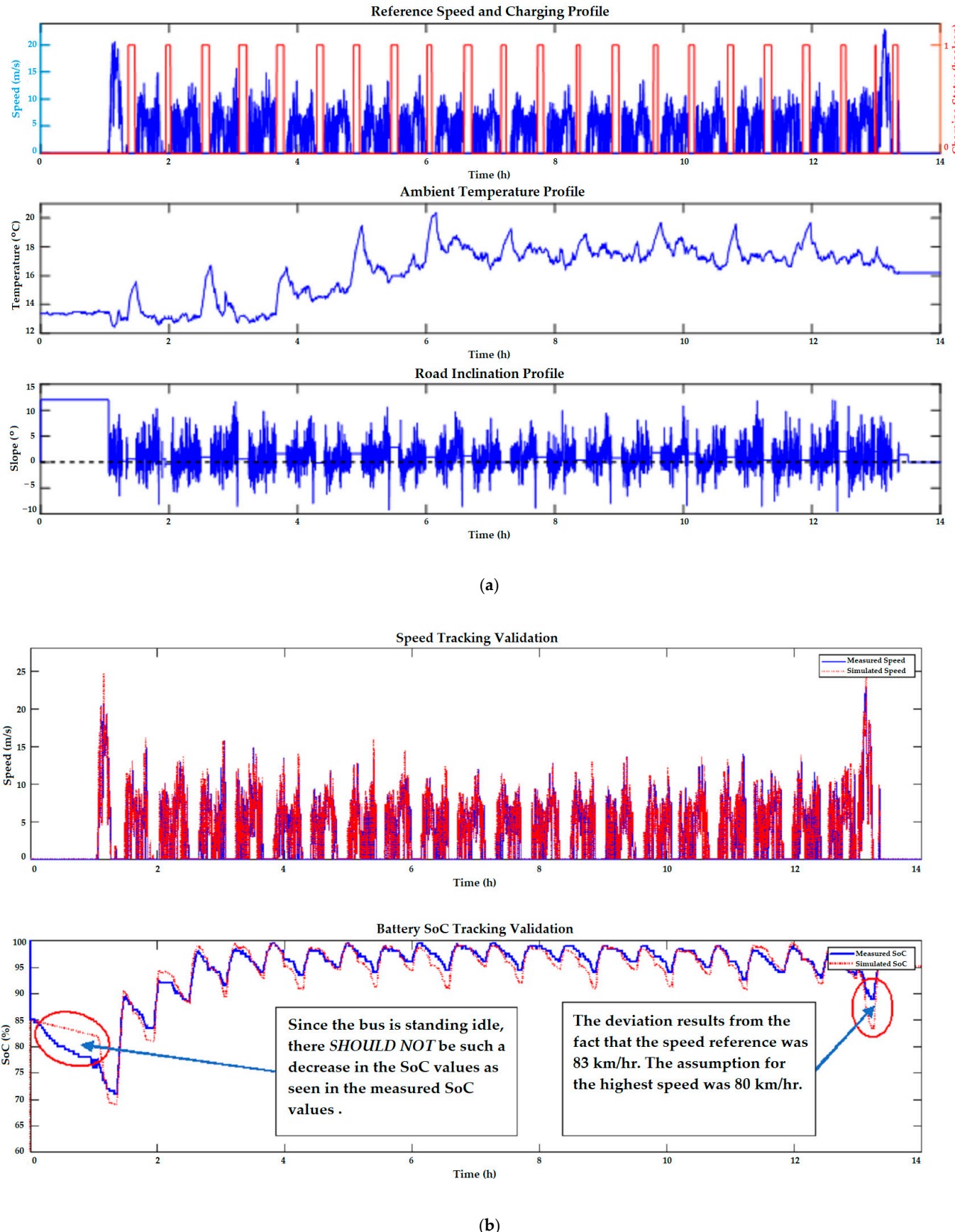

**Figure 10.** Validation of the May 27th demonstration of 12 m bus in Gothenburg city using the estimates in Table 6. (**a**) Scenario inputs, (**b**) Scenario outputs and validation.

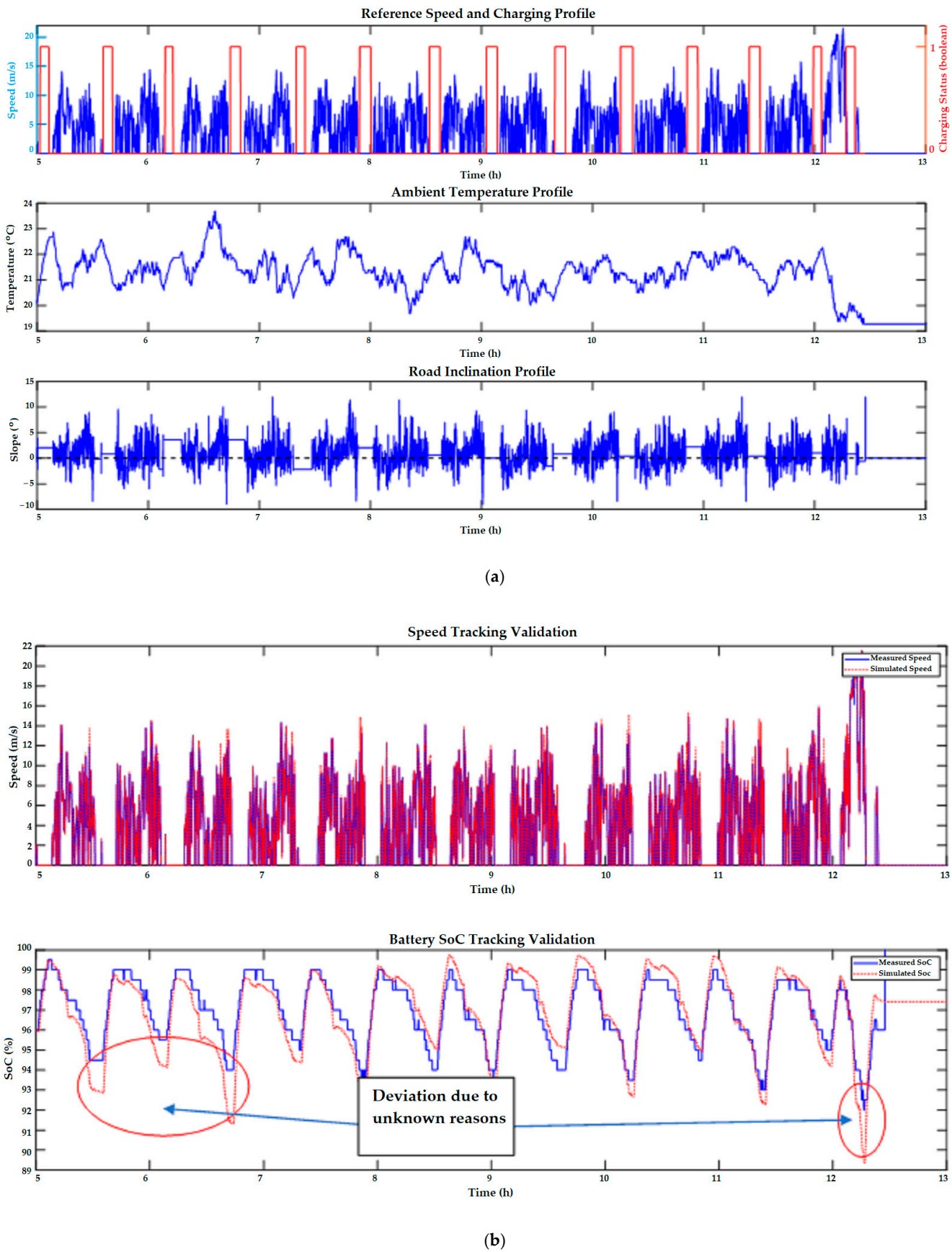

**Figure 11.** Validation of the May 29th demonstration of 12 m bus in Gothenburg city using the estimates in Table 6. (**a**) Scenario inputs, (**b**) Scenario outputs and validation.

## 6. Conclusions

This study presents a methodology for improving the accuracy of a Lo-Fi model of the electric bus powertrain using measurement data from 12 m and 18 m electric bus demonstrations in cities. First, a qualitative comparison is made of the bus's energy requirements between the baseline UC simulations, which used a standard driving profile, and the actual driving profile from the demonstrations. The results show that in Barcelona, the energy requirements of the 12 m buses were 17.5% higher, while those of the 18 m buses were 33% higher, when using the driving profile of the demonstration. For Osnabruck, the energy requirements were 20% lower for the 12 m buses when using the driving profile of the demonstrations, while the 18 m buses had similar energy requirements to the baseline. This is because the Barcelona demonstrations had a higher average velocity compared with the baseline, while the Osnabruck 12 m bus demonstrations had a lower average velocity. The magnitude of the acceleration and deceleration had less effect on the energy requirements of an electric powertrain, since energy expended during accelerations are recovered during decelerations. Only in cases where the driving profile showed many hard decelerations did the energy requirement become higher; this was because during hard decelerations, the bus requires friction brakes to decelerate in addition to the electric motor, leading to less energy recovered via regeneration.

Next, the measurement data of the vehicle's speed profile from the demonstration were used as inputs to the simulation framework, and the simulation results of the battery SoC profile were compared to measured battery SoC profiles from the demonstrations. A tuning methodology, based on iSOPT optimization, combined with splitting the simulation into smaller time windows during optimization, was used to minimize the NRMSE between the simulated and measured battery SoC signals and ensure that there is a high degree of correlation between them. The results show that the tuning process based on the window technique applied to the optimization process successfully synchronized the simulation and measurement outputs quicker than the technique presented in [5]. In rare cases, deviations are encountered between the simulated and measured output. Of these, the deviations that describe a situation that is physically impossible, based on the data provided, are ignored. Other deviations result from limitations in the assumptions made during the design of the simulation framework, and those were fixed by correcting the assumptions. However, in two cases, deviations occurred for which no suitable explanation could be determined, and those would require further research to fix. Overall, the optimization achieved more than 90% correlation between the simulated and measured SoC profile.

The techniques utilized in this research will be refined further in future research to perform real-time tuning of the platform with the aim of deploying a cloud-based DT of the electric bus that will be able to make predictions in real-time based on the measurement data from the real vehicle. To achieve that goal requires two systems working in synergy: first, it would be necessary to invest in CAN dataloggers with WiFi or 3G/4G capability that will capture the sensor data from the vehicle's CAN network and periodically transmit these measurements to a cloud server. Then, a highspeed simulation model needs to be deployed in the cloud server that will periodically take in these measurements data as inputs and quickly simulate the outputs and tune itself using appropriate tuning techniques to minimize the error between the simulated and measured outputs. The key will be to reduce the simulation time needed during the tuning process (whether via machine learning or optimization), so the model can tune itself in real time. This requires further improvements to the optimization technique and utilizing machine learning using artificial neural networks. Machine learning algorithms are also able to adapt to changes in behavior over time. Once the error has been reduced below an acceptable threshold, then many virtual copies of the DT can be deployed in the cloud to act as virtual testbeds for a myriad of different tests, or to simulate fleets of such vehicles to investigate the charging infrastructure requirements in city bus routes and depots.

**Author Contributions:** Conceptualization, M.M.H. and N.A.; methodology, M.M.H., N.A. and M.R.; software, M.M.H. and N.A.; validation, M.M.H.; formal analysis, M.M.H.; investigation, M.M.H. and N.A.; resources, N.A.; data curation, N.A.; writing—original draft preparation, M.M.H.; writing—review and editing, N.A., M.R., M.E.B. and O.H.; visualization, M.E.B. and O.H.; supervision, O.H.; project administration, M.E.B. and O.H.; funding acquisition, O.H. All authors have read and agreed to the published version of the manuscript.

**Funding:** This research was funded by the European Commission—Innovation and Networks Executive Agency, Grant number 769850, under the title of ASSURED—H2020-GV-2016-2017/H2020-GV-2017.

**Data Availability Statement:** Data used in this article are private and can be found in the project deliverables for those having access.

**Acknowledgments:** The authors acknowledge Flanders Make for their support to this research group. The authors acknowledge the OEMs and CBOs for providing measurement data from demonstrations for analysis in this article.

**Conflicts of Interest:** The authors declare no conflict of interest. The funders had no role in the design of the study; collection, analyses, or interpretation of data; writing of the manuscript; or decision to publish the results.

## Appendix A. Input Scenario for Bus Demonstrations

*Appendix A.1. Barcelona City*

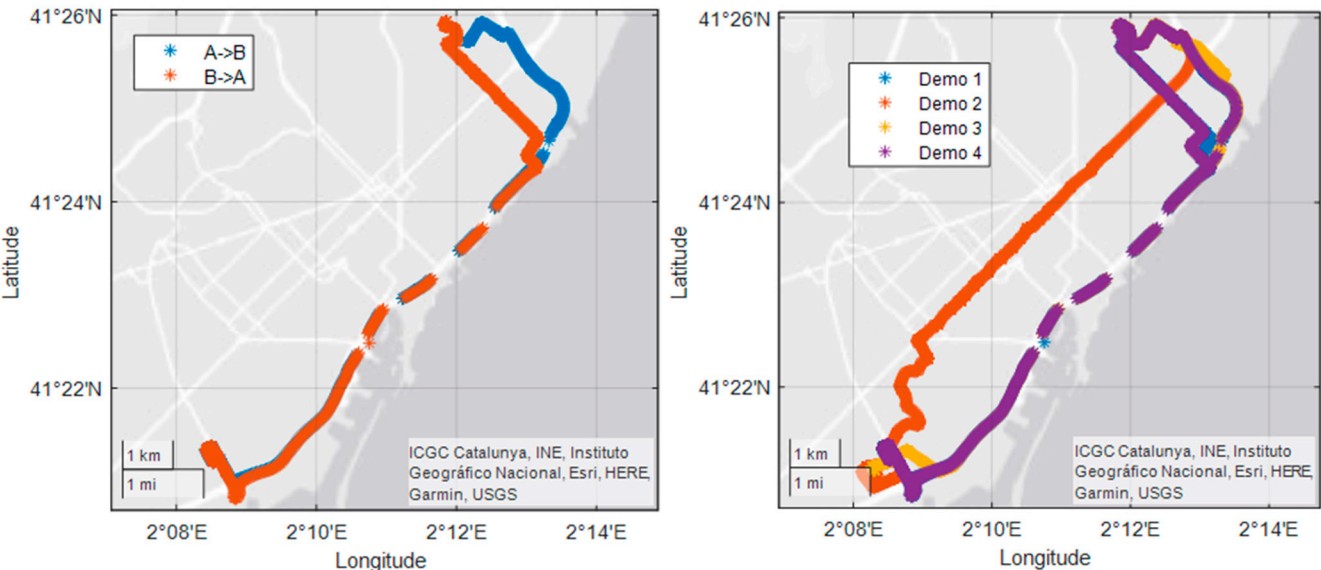

**Figure A1.** Route map of H16 for the demonstration.

*Appendix A.2. Osnabruck City*

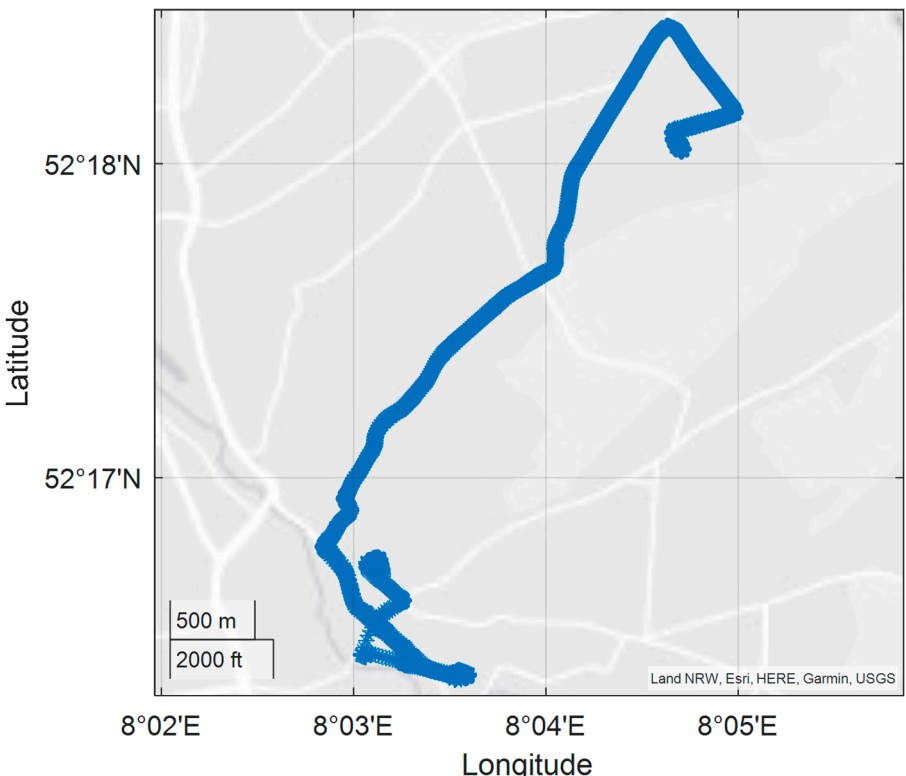

**Figure A2.** Route map of N5 for the demonstration.

*Appendix A.3. Gothenburg City*

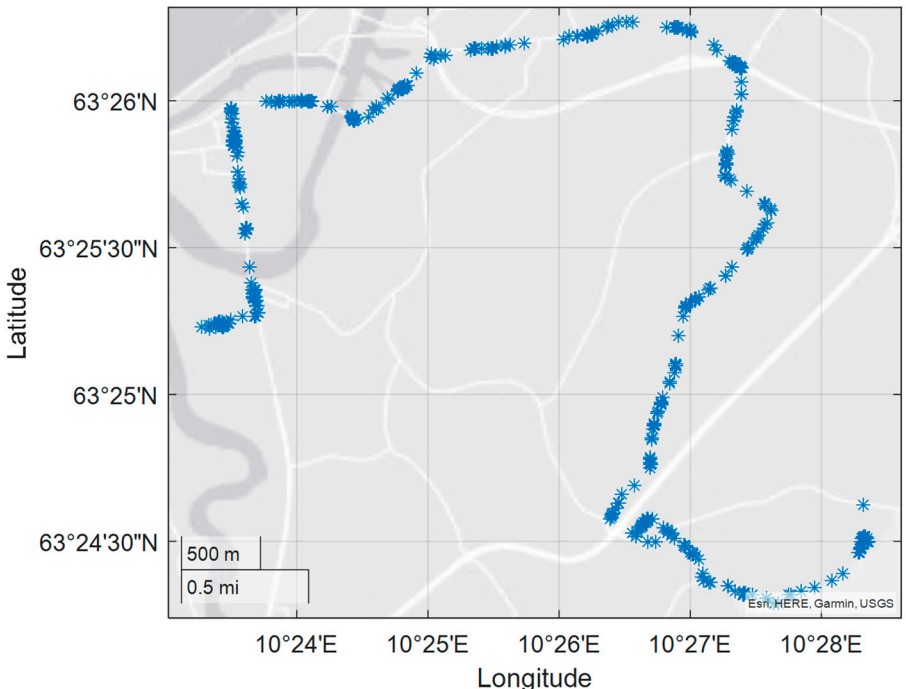

**Figure A3.** Route map of R55 for the demonstration.

## Appendix B. Review of Energy Management (ECO) Features

*Appendix B.1. ECO-Driving Functionality*

ECO-driving transforms the driving cycle into an eco-friendlier profile that limits the maximum acceleration and speed of the vehicle resulting in less tractive energy requirements; furthermore, it also optimizes the energy recovery during regeneration by keeping the EM in the optimum power band to recover the maximum power. As can be seen from Figure A4, the velocity profile is smoothened by application of a ramp to the acceleration. The velocity modification ensures smoother changes in velocity and removes discontinuity in the acceleration. The top velocity and acceleration are also limited to save energy. The overall driving behavior is gentler, with minimal hard accelerations and braking. This is important because, unlike normal braking action, hard braking is not as efficient at energy recovery as a large portion of the braking power needs to be diverted to the friction brakes, rather than the electric motor, to cope with the braking load. This is why applying ECO-driving to an aggressive driving style results in significant energy savings. Therefore, good driving behavior is a requirement for proper regenerative braking action and is a core component of ECO-driving. The ECO-driving method also ensures that regardless of the velocity modification, the distances traveled between the ECO and non-ECO version remains synchronized. This distance synchronization is important to convince many CBOs to adopt ECO-driving principles for their routes, as they can still maintain their default bus schedules even while limiting top speed and acceleration.

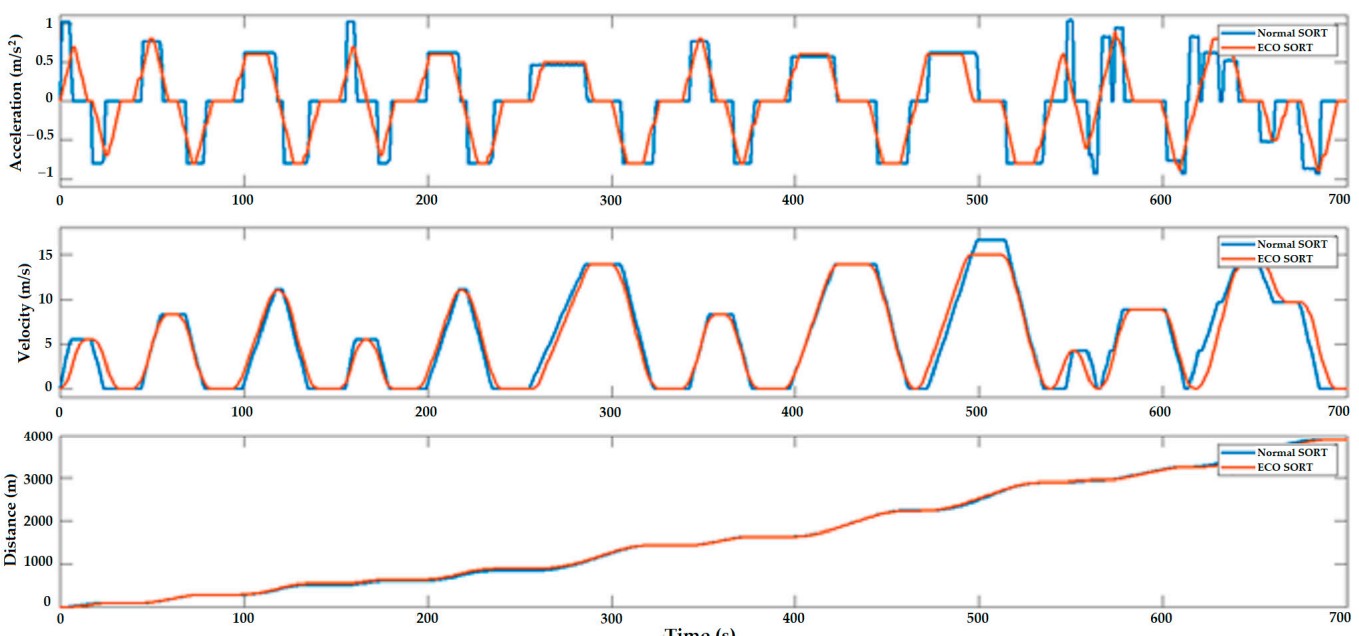

**Figure A4.** Velocity modification for Eco-friendly profile generated for a standard driving cycle.

*Appendix B.2. ECO-Comfort Functionality*

Figure A5 shows how the ECO-comfort functionality dynamically alters the cabin setpoint temperature throughout the day. The dynamic temperature setpoint of the ECO-comfort depends on the passenger count inside the bus as well as the ambient temperature. The temperature setpoint is devised to save the energy required for climate control at the expense of slightly reduced passenger comfort. This means a little less cooling inside the bus during summers and a little less heating inside the bus during winters. As well as dynamic temperature setpoints, ECO-comfort also uses pre-conditioning to reduce the energy requirement needed for heating or cooling when the bus is in motion. Pre-conditioning means to utilize the thermal management system to track the setpoint temperature of the bus while it is connected to the grid for charging; thus appropriating the energy from the

grid instead of the battery. The energy reduction by ECO-comfort is highly dependent on the climate, e.g., for a hot climate, the maximum energy reduction due to ECO-comfort is achieved during mid-summer, while for colder climates, the maximum energy reduction is attained in mid-winter.

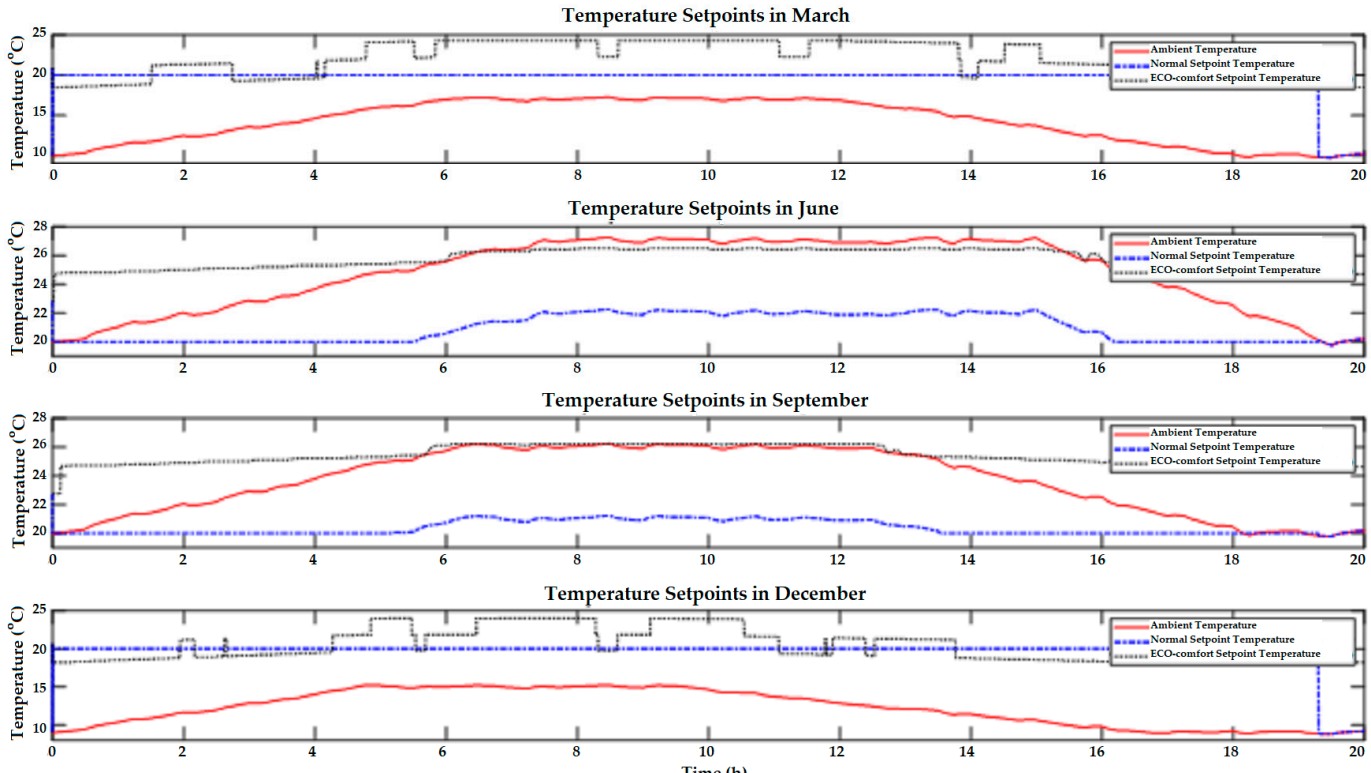

**Figure A5.** The daily dynamic cabin setpoint temperature for a 12 m bus.

*Appendix B.3. ECO-Comfort Functionality*

Figure A6 shows the ECO-charging functionality, which makes use of pulsed charging, instead of continuous charging. Since the charging is pulsed, the battery has a chance to cool down in between the charging pulses; this reduces the temperature increase during charging, and necessitates less cooling by the HVAC system. At the same time, this also results in low c-rate charging on average, thus improving battery longevity. The disadvantage of this charging method is that the battery will take longer to charge; to mitigate this, either the charging duration needs to be increased, which is not always possible due to bus scheduling constraints, or the battery size needs to be increased so that the battery can deliver the range required during its scheduled operational period. Thus, ECO-charging prevents excessive battery heating during charging, has minimal effect on the vehicle's energy requirements, and lowers the load on the electricity grid.

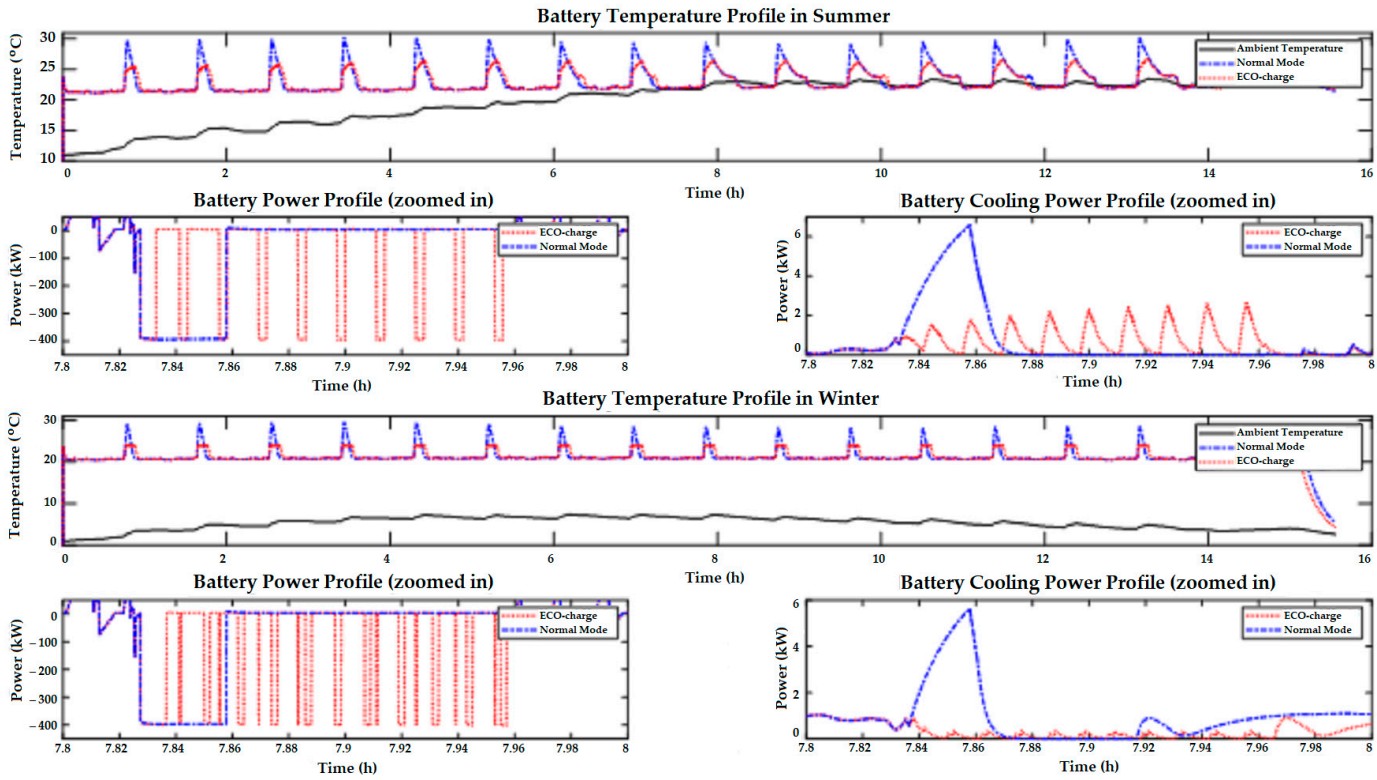

**Figure A6.** ECO-charging profile highlighting the effects of pulsed charging functionality.

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
