# Peer review of "Parameter Optimization and Tuning Methodology for a Scalable E-Bus Fleet Simulation Framework: Verification Using Real-World Data from Case Studies"

_applsci, doi:10.3390/app13020940_

Round 1

Reviewer 1 Report

The work entitled "Parameter Optimization and Tuning Methodology for a Scalable e-Bus Fleet Simulation Framework: Verification using Real-world Data from Case Studies" touches on a very interesting topic. In the introduction, the authors describe computer-aided automotive systems, but the introduction contains too much historical data that does not contribute anything to the work. The manuscript is very long, it is 33 pages long and in my opinion it should be shortened. The description of the simulation framework is very extensive, which makes it chaotic, it is difficult to verify which of the presented information is the most important. The simulation steps taken should be more detailed, they are currently hidden in the text, they should be emphasized and presented in order. Figure 2 doesn't add much to the work. Each of the presented graphs should be briefly characterized and its justification for publication should be explained. The work, despite small errors, is at a very good level and very interestingly written.

Author Response

Dear reviewer,

I wish to thank you profusely for your thoughtful comments. They made me look at my article with a more critical air and reevaluate its contents. I hope the edits I make here can, even partially, allay your concerns regarding my paper.

Comment:

In the introduction, the authors describe computer-aided automotive systems, but the introduction contains too much historical data that does not contribute anything to the work.

Response:

When I decided on the introduction, I wanted to approach the narrative from a different angle. Typically, in papers with electric vehicles, the narrative starts in an all-too-common format (global warming, need to reduce greenhouse gas emissions etc.) and it tends to get stale. Therefore, I wished to change it and provide a different perspective. I thought about what the research presented in the paper could address, even peripherally, and it looked like the research could be the precursor to digital twins. Since, DTs are more associated with the manufacturing industry, I started my introduction with this frame of mind. However, taking your concerns into consideration, I have removed 7.5 lines from the first paragraph (a considerable amount of historical data in fact), without unduly changing the narrative I wished to present.

Comment:

The manuscript is very long, it is 33 pages long and in my opinion it should be shortened.

Response:

I whole heartedly agree with this point. Such a long paper was also difficult to write from our end. However, our paper is only 23 pages long. The other 10 pages are the Appendix, which is optional for the reader. If a reader does not wish to read the appendix, there is absolutely no loss to the reader, because all the important information needed to understand our research are contained within the body of the article. I have corrected this by shortening the appendix considerably and removing all unnecessary materials. Furthermore, I applied the same process to sections 3 and 4 that was applied to section 2 (see response to next comment) and discarded repeated ideas/points. With these edits, I have managed to shorten the paper by 6 pages.

Comment:

The description of the simulation framework is very extensive, which makes it chaotic, it is difficult to verify which of the presented information is the most important.

Response:

I have gone through the section carefully have managed to eliminate a full 20 lines of description from this section. I have specifically looked for and eliminated repetitions of the same idea. Now the section is leaner and essentially mentions 4 key points in as many paragraphs: brief intro, description of the model/framework, the data collection process, and the data processing part. It is my sincere hope now the average reader will not get confused while reading this section.

Comment:

The simulation steps taken should be more detailed, they are currently hidden in the text, they should be emphasized and presented in order.

Response:

The simulation steps start in section two with data collection and pre-processing. It then continues in section 4, with the description of the optimization and tuning methodology. Finally, it ends with a brief mention in section 5, where the way the differences in collected data between the two cities was resolved within the simulation. Unfortunately, I see no was of combining them into one section.

Comment:

Figure 2 doesn't add much to the work. Each of the presented graphs should be briefly characterized and its justification for publication should be explained.

Response:

Figure 2 has been removed, and subsequent figure numberings have been adjusted. For, the remaining part of this response please assume the new figure numberings.

As for the justification of the figures presented in the article, Figures 3 ~ 5 and 8 ~ 11 are results and are required in this article to explain certain conclusions presented in the text. Figures 2, 6, 7, 9 ~ 11 are inputs to the simulation and also required, as the simulation outputs are explained in terms of the input used. Figure 1is an overview of the electric bus powertrain, and enhances the description given in section 2, so it is necessary.

Reviewer 2 Report

This paper presents the methodology of improving the accuracy of a Lo-Fi model of the electric bus powertrain using measurement data from 12m and 18m electric bus demonstrations in cities. The proposed approach shows good and reasonable results. Overall, the quality is fine, except for minor issues:

1.Line 137, "3.1. Inputs to the Simulation Framework", should be "2.1".

2.Authors should explain Eqn.1 .

Author Response

Dear reviewer,

Thank you for your suggestions. I have taken care of all of them.

Comment:

1.Line 137, "3.1. Inputs to the Simulation Framework", should be "2.1".

Response:

Thank you for catching this error. The section number has been corrected, and further review of all section numbering also led us to correct section 4.2, which was mistakenly numbered 3.1.

2.Authors should explain Eqn.1 .

We have explained equation 1 by adding the complete formula and then explaining the effect of the formula. Please see lines: 304 ~ 316.

Reviewer 3 Report

1)     Explain in detail the reasons for the research contribution or technical impact score. What is the new contribution of this paper?

2)     The authors have to highlight their contribution in a better manner.

3)      Some figures must be improved

4)     Kindly, improve the presentation of your findings in the conclusion

Author Response

Dear reviewer,

I thank you for taking the time to review my paper, and I appreciate your thoughtful insights. I hope I can address your concerns to your satisfaction.

Comment:

  • Explain in detail the reasons for the research contribution or technical impact score. What is the new contribution of this paper?

Response:

I have added a clear objective to the introduction regarding the ‘why’ of this research (lines 80 ~ 83). Next, in the end of section 4, I have added a paragraph regarding the novelty of this research work, over the one conducted previously (lines 398 ~ 413). Finally, in the conclusion, I have described the future work that will be done using the techniques deployed in this research (lines 589 ~ 606). I hope these three parts provide a compelling narrative for a reader.

Comment:

  • The authors must highlight their contribution in a better manner.

Response:

I have made extensive modifications to the article and removed many unnecessary details. Thus, the remaining text explains more clearly the methodology. Also, as part of the edits made for comment 1, the contribution of this paper has been highlighted.

Comment:

  • Some figures must be improved

Response:

I have removed a figure that served no purpose (figure 2 in the original draft) and edited figure 7 (in the original draft, but now figure 6) to explain the methodology better. As for the differences in the figure styles, it is due to some figures being generated from Microsoft Excel, while others were generated from MATLAB. I apologize for this.

Comment:

4)     Kindly, improve the presentation of your findings in the conclusion

Response:

I have made the conclusion more comprehensive and split it into three parts, one summarizes the comparison between the demo and the UC simulations through an empirical approach. The 2nd part summarizes the new optimization technique used, and the last part gives details of the future work to be done in this research track

Reviewer 4 Report

Page 4. Table 1. Why were some parameters measured in only one city, eg ambient temperature in Gothenburg?

Page 5. Point 3. For what purpose are the names of cities repeated once again?

Page 6. Table 3. Why was the speed unit km/h (Table 1) changed to m/s? The same remark applies to Figure 3.

Page 6. Table 3. Please use superscript [m/s2].

Page 18. Table 6. Please use superscript [oC].

Page 33. Item 17. Please eliminate unnecessary spaces.

Author Response

Dear reviewer,

Thank you for your queries. I hope that I have addressed them to your satisfaction:

Comment:

Page 4. Table 1. Why were some parameters measured in only one city, eg ambient temperature in Gothenburg?

Response:

The arbitrary nature of the data collection process was due to different stakeholders that were responsible for the data collection process. Some were more willing to share data than others. The only requirement we imposed upon them were:

  • That the data must contain these two measurement values:
    • Vehicle speed
    • Battery SoC
  • The minimum data logging frequency was 1Hz.

Any extra measurement data provided or measurement data with a higher sampling rate (than 1Hz) were up to their discretion. (NOTE: Even so, as you can see from Table 1, some stakeholders under sampled their measurement data).

I have re-edited the paragraph above Table 1, to make this clearer. (Lines 122 ~ 136)

Comment:

Page 5. Point 3. For what purpose are the names of cities repeated once again?

Reviewer:

I assume that you are referring to the start of section 3. With that assumption, I have rephrased the sentence structure, so it does not feel like I am repeating myself. I hope this is suitable.

Comment:

Page 6. Table 3. Why was the speed unit km/h (Table 1) changed to m/s? The same remark applies to Figure 3.

Response:

The authors thank you for catching this oversight on our part. The speed values in Table 3 are now correctly presented in kmph. The reason Table 3 mentioned m/s was because the authors directly took m/s values from the simulation (since the simulation was run using the SI unit of speed – m/s). The top plot of figure 3 has also been changed appropriately

Comment:

Page 6. Table 3. Please use superscript [m/s2].

Response:

This has been corrected in Table 3

Comment

Page 18. Table 6. Please use superscript [oC].

Response:

This has been corrected in Table 6.

Comment:

Page 33. Item 17. Please eliminate unnecessary spaces.

Response:

The space due to the tab stop has been removed

Round 2

Reviewer 3 Report

 Accept in present form